# Inflammation rapidly recruits mammalian GMP and MDP from bone marrow into regional lymphatics

Juana Serrano-Lopez[1†], Shailaja Hegde[1,2†], Sachin Kumar[1], Josefina Serrano[3], Jing Fang[1], Ashley M Wellendorf[1], Paul A Roche[4,5], Yamileth Rangel[3], Leolene J Carrington[6], Hartmut Geiger[1,7], H Leighton Grimes[8], Sanjiv Luther[9], Ivan Maillard[6], Joaquin Sanchez-Garcia[3], Daniel T Starczynowski[1,10], Jose A Cancelas[1,2]*

[1]Divisions of Experimental Hematology, Cincinnati Children's Hospital Medical Center, University of Cincinnati College of Medicine, Cincinnati, United States; [2]Hoxworth Blood Center, University of Cincinnati College of Medicine, Cincinnati, United States; [3]Hematology Department, Reina Sofía University Hospital/Maimonides Biomedical Research Institute of Córdoba (IMIBIC)/University of Córdoba, Córdoba, Spain; [4]Center for Cancer Research, National Cancer Institute, Bethesda, United States; [5]Experimental Immunology Branch, National Cancer Institute, National Institutes of Health, Bethesda, United States; [6]University of Pennsylvania Perelman School of Medicine, Philadelphia, United States; [7]Institute of Molecular Medicine, Ulm University, Ulm, Germany; [8]Immunobiology, Department of Pediatrics, Cincinnati Children's Hospital Medical Center, University of Cincinnati College of Medicine, Cincinnati, United States; [9]Center for Immunity and Infection, Department of Biochemistry, University of Lausanne, Epalinges, Switzerland; [10]Department of Cancer Biology, University of Cincinnati, Cincinnati, United States

*For correspondence:
jose.cancelas@uc.edu

[†]These authors contributed equally to this work

Competing interests: The authors declare that no competing interests exist.

**Abstract** Innate immune cellular effectors are actively consumed during systemic inflammation, but the systemic traffic and the mechanisms that support their replenishment remain unknown. Here, we demonstrate that acute systemic inflammation induces the emergent activation of a previously unrecognized system of rapid migration of granulocyte-macrophage progenitors and committed macrophage-dendritic progenitors, but not other progenitors or stem cells, from bone marrow (BM) to regional lymphatic capillaries. The progenitor traffic to the systemic lymphatic circulation is mediated by Ccl19/Ccr7 and is NF-κB independent, Traf6/IκB-kinase/SNAP23 activation dependent, and is responsible for the secretion of pre-stored Ccl19 by a subpopulation of CD205+/CD172a+ conventional dendritic cells type 2 and upregulation of BM myeloid progenitor Ccr7 signaling. Mature myeloid Traf6 signaling is anti-inflammatory and necessary for lymph node myeloid cell development. This report unveils the existence and the mechanistic basis of a very early direct traffic of myeloid progenitors from BM to lymphatics during inflammation.

## Introduction

Bacterial infections represent one of the major threats for the human immune system and can lead to sepsis and death (*Martin et al., 2003*). A functional immune response is a key factor to control the outcome of bacterial infections. Therefore, the human immune system has evolved several effector mechanisms to fight bacterial infections that involve the innate and the adaptive arms of the immune system. Antigen presentation is an essential mechanism of activation that requires crosstalk

**eLife digest** When the body becomes infected with disease-causing pathogens, such as bacteria, the immune system activates various mechanisms which help to fight off the infection. One of the immune system's first lines of defense is to launch an inflammatory response that helps remove the pathogen and recruit other immune cells. However, this response can become overactivated, leading to severe inflammatory conditions that damage healthy cells and tissues.

A second group of cells counteract this over inflammation and are different to the ones involved in the early inflammatory response. Both types of cells – inflammatory and anti-inflammatory – develop from committed progenitors, which, unlike stem cells, are already destined to become a certain type of cell. These committed progenitors reside in the bone marrow and then rapidly travel to secondary lymphoid organs, such as the lymph nodes, where they mature into functioning immune cells. During this journey, committed progenitors pass from the bone marrow to the lymphatic vessels that connect up the different secondary lymphoid organs, and then spread to all tissues in the body. Yet, it is not fully understood what exact route these cells take and what guides them towards these lymphatic tissues during inflammation.

To investigate this, Serrano-Lopez, Hegde et al. used a combination of techniques to examine the migration of progenitor cells in mice that had been treated with lethal doses of a bacterial product that triggers inflammation. This revealed that as early as one to three hours after the onset of infection, progenitor cells were already starting to travel from the bone marrow towards lymphatic vessels. Serrano-Lopez, Hegde et al. found that a chemical released by an "alarm" immune cell already residing in secondary lymphoid organs attracted these progenitor cells towards the lymphatic tissue.

Further experiments showed that the progenitor cells travelling to secondary lymphoid organs were already activated by bacterial products. They then follow the chemical released by alarm immune cells ready to respond to the immune challenge and suppress inflammation. These committed progenitors were also found in the inflamed lymph nodes of patients.

These findings suggest this rapid circulation of progenitors is a mechanism of defense that contributes to the fight against severe inflammation. Altering how these cells migrate from the bone marrow to secondary lymphoid organs could provide a more effective treatment for inflammatory conditions and severe infections. However, these approaches would need to be tested further in the laboratory and in clinical trials.

between the innate and adaptive immune system to fight bacterial infections. Dendritic cells are short-lived professional antigen-presenting cells (APCs), and their life span is further reduced during the inflammatory response to pathogens (*Kamath et al., 2002*). Upon inflammation, primed APC thus need to be replaced.

During inflammation, systemic signals alert and activate bone marrow (BM) hematopoietic stem cells and progenitors (HSC/P) (*Chavakis et al., 2019*; *Mitroulis et al., 2018*; *Nagai et al., 2006*; *Ueda et al., 2005*). Inflamed secondary lymphoid organs such as lymph nodes (LN) recruit antigen-presenting dendritic cells (DCs) (*Legler et al., 1998*; *Luther et al., 2000*; *Saeki et al., 1999*), while pathogen-associated molecular pattern signals (PAMPs) trigger migration of tissue-resident DC to the LN (*Kaisho and Akira, 2001*; *Sallusto and Lanzavecchia, 2000*). Circulation of HSC/P that enter the lymphatic vessels from the peripheral blood (PB) with ability to amplify APCs has been described (*Massberg et al., 2007*). However, the circuits used by these HSC/P populations, their characterization, and the cellular and molecular mechanisms that regulate this traffic in inflammatory conditions have not been addressed in detail.

Lymphatics form part of an open circulatory system that drains cells and interstitial fluid from tissues. Recently, bone lymphatic endothelial cells have been shown to arise rapidly from pre-existing regional lymphatics in inducible bone-expressing *Vegfc* transgenic mice through Vegfr3, osteoclast activation, and bone loss (*Hominick et al., 2018*; *Monroy et al., 2020*). Acute endotoxemia is associated with osteoclast activation and bone loss (*Hardy and Cooper, 2009*; *Nason et al., 2009*). We postulated the pre-existence of an anatomical and functional patent circuit that communicates BM and lymphatic tissues that can be induced upon severe inflammatory conditions like endotoxemia.

Our work identifies an emergent traffic of DC-biased myeloid progenitors through direct transit from BM to bone lymphatic capillaries. This traffic is highly activated in endotoxic inflammation. In human reactive lymphadenitis or just after a single immune endotoxic challenge, such as following lipopolysaccharide (LPS) stimulation in mice, a massive mobilization of myeloid progenitors from the BM to lymph and retention in the LN takes place. The mobilization is rapid prior to their appearance in PB. LPS simultaneously induces cell-autonomous Ccr7 expression on granulocyte-macrophage progenitors (GMPs) and macrophage-dendritic progenitors (MDPs), and a non-cell-autonomous myeloid cell-dependent secretion of Ccl19 in the LN. In vivo blockade of LPS signaling in mature myeloid cells, deletion of hematopoietic Ccl19, or neutralization of Ccr7 completely abrogated the GMP/MDP migration from the BM to the LN. Moreover, genetic and pharmacological approaches revealed that Traf6-Irak1/4-Ubc13-IκB kinase (IKK) signaling mediates NF-κB-independent-SNAP23 phosphorylation and secretion of pre-formed Ccl19 from a specific population of conventional dendritic cells (cDCs), and mature myeloid cell Traf6-dependent signaling is of anti-inflammatory nature. These findings indicate that inflammation results in mobilization of cDC-forming cells directly from the BM to the lymph and LN. As such, emergent myeloid lineage mobilization from the BM to lymph may be important in inflammation by acutely differentiating into antigen-presenting precursors in lymph tissues and associate with an anti-inflammatory response in endotoxemia.

## Results

### Inflammation associates with emergent migration of myeloid progenitors, but not HSC, from BM to lymphatics

To determine whether there is a circulation of HSC/P to human LN, we prospectively analyzed the presence of side population (SP) cells in LN biopsies (*Figure 1—figure supplement 1A*) obtained from lymphadenitis and lymphoma patients at diagnosis. Human and murine SP cells, with ability to extrude the dye Hoechst 33342 through upregulated activity of multidrug resistance protein complexes (*Zhou et al., 2001*) in BM and other tissues (*Brusnahan et al., 2010*; *Challen and Little, 2006*; *Goodell et al., 1996*), are enriched in long-term reconstituting HSC and other more committed populations of progenitors (*Matsuzaki et al., 2004*; *Weksberg et al., 2008*). We found an SP population at a frequency higher than 0.01% in 36 out of 64 LN biopsies (53.12%). However, the content of SP cells in the LN did correlate with the LN histological diagnosis. The elevated frequency of SP cells in LN did correlate with the LN histological diagnosis (*Figure 1A*) but not to the anatomical location of the lymphadenopathy (*Figure 1—figure supplement 1B* and *Supplementary file 1*). The accumulation of SP cells was significantly higher in LN from lymphadenitis patients than in lymphoma patients. Further dissection based on histological classifications by independent pathology analysis resulted in the lymphadenitis specimens being sorted into distinct histological categories that corresponded to follicular lymphadenitis with paracortical predominance (FL), granulomatous lymphadenitis (GL), and lymphadenopathies with histological or molecular evidence of viral etiology (viral lymphadenitis [VL]). Interestingly, FL and GL LN contained a median of 0.2% SP cells with a range from <0.01% to ~40%, which was significantly higher than the content of SP cells in VL, Hodgkin's lymphoma, and non-Hodgkin's lymphoma LN (*Figure 1A*, *Figure 1—figure supplement 1A* and *Supplementary file 1*). The existence of myeloid-committed hematopoietic progenitors was confirmed in myeloid colony-forming cell unit (CFU) assays (*Figure 1—figure supplement 1C*) performed on samples from patients with FL. These data show that non-viral inflammatory lymphadenitis results in a significantly increased frequency of primitive hematopoietic cells in LN, while it does not reveal the type of progenitor cells. To confirm whether LN SP cells indeed contained HSC/P, we first sorted LN SP cells from patients with reactive lymphadenitis and plated them in methylcellulose cultures containing rhIL-3, rhIL-6, and rhSCF cytokines. CFU analysis demonstrated that SP cells were indeed capable of producing myeloid colonies (*Figure 1—figure supplement 1D*), while non-SP cells were devoid of measurable CFU-forming ability (data not shown). Immunophenotypic analysis of SP-derived progenitors was also consistent with enrichment of a heterocellular population of CD34 and CD133 expressing granulocyte-, granulocyte-macrophage-, and cDC-biased progenitors (*Bornhäuser et al., 2005*; *Görgens et al., 2013*; *Figure 1—figure supplement 1D*). The vast majority of CD45$^+$/CD34$^+$ cells co-expressed CD133$^+$, and the CD45$^+$/CD34$^-$ population was split ~50:50 into CD133$^+$ and CD133$^-$ cells (*Figure 1—figure supplement 1D*). In combination, these data show

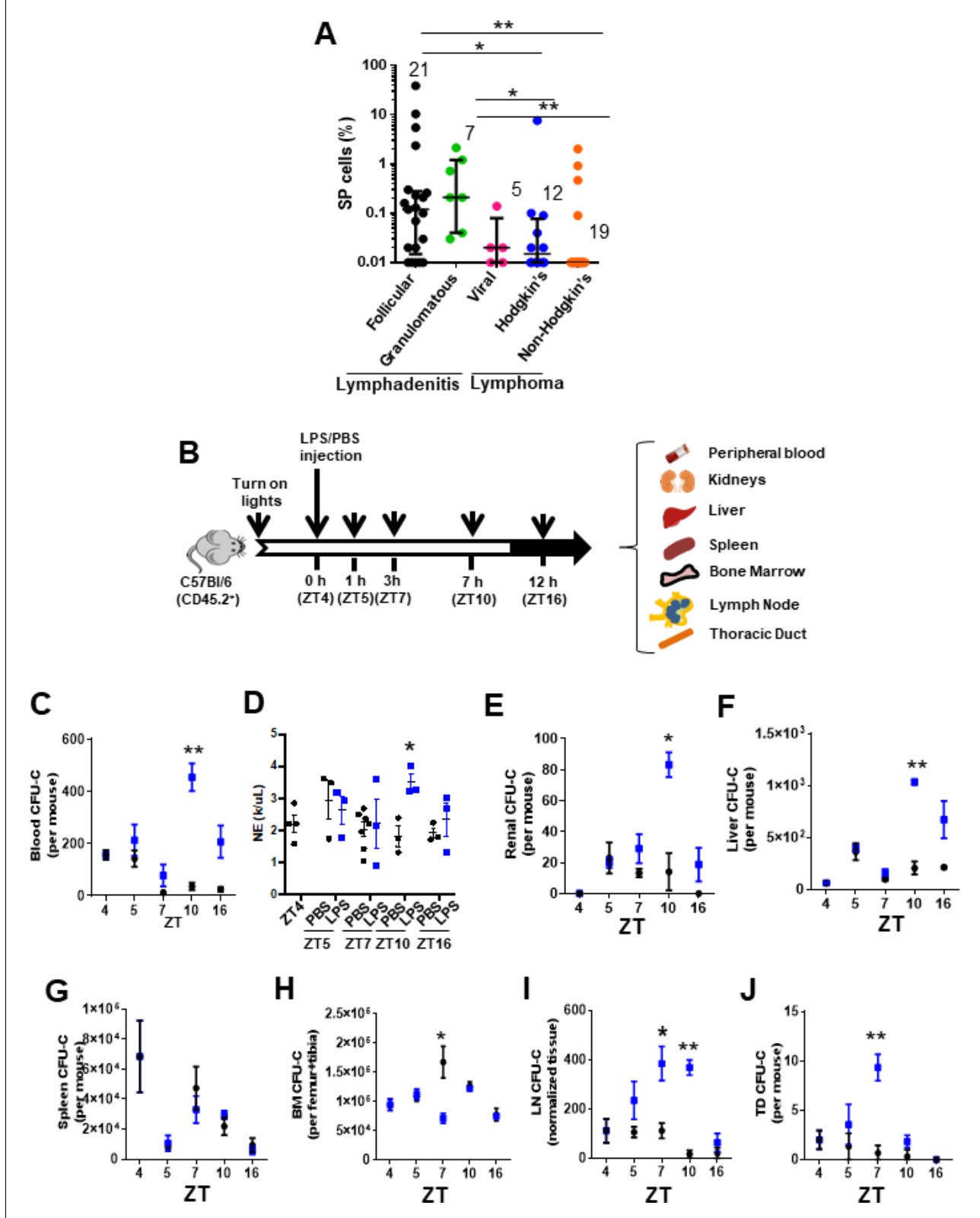

**Figure 1.** Inflammation induces early mobilization of hematopoietic stem cells and progenitors to lymph organs in humans and mice. (**A**) Content of side population (SP)cells in human lymph node (LN) by flow cytometry. LN biopsies had been blindly identified histologically as lymphadenitis, subcategorized in follicular (FL, black circles, n = 21), granulomatous (GL, green circles, N = 7), and viral (VL, pink circles, n = 5) and lymphomas, subcategorized in Hodgkin's lymphoma (HL, blue circles, n = 12) or non-Hodgkin's lymphoma (NHL, orange circles, n = 19). (**B**) Strategy for

*Figure 1 continued on next page*

*Figure 1 continued*

lipopolysaccharide (LPS) administration and collection of tissues (blood, kidneys, liver, spleen, bone marrow (BM), lymph node (LN), and thoracic duct (TD)) at specific times. LPS or vehicle control PBS was administered at the early rest phase into C57Bl/6 (CD45.2[+]) mice, and tissue specimens were collected before (zeitgeber time [ZT]4) 1 hr (ZT5), 3 hr (ZT7), 6 hr (ZT10), or 12 hr (ZT16) later. (C) Myeloid colony-forming-cell unit (CFU-C) content in peripheral blood (PB) from C57Bl/6 mice pre-treated with PBS (black circles) or LPS (blue circles) at different circadian cycle times. (D) Absolute neutrophil count in PB from C57Bl/6 mice pre-treated with PBS (black circles) or LPS (blue circles) at different circadian cycle times. (E–J) Myeloid CFU-C content in organs at different circadian cycle times. CFU-C contained in kidneys (E), liver (F), spleen (G), BM (H), LN (I), and TD (J) in response to PBS (black circles) or LPS (blue squares) at different circadian cycle times (n = 3–4 mice per time point and treatment). ZT4 = 10 am (time of LPS administration). ZT7: 1 pm (3 hr post-administration of LPS). ZT10: 4 pm (6 hr post-administration of LPS). ZT16: 10 pm (12 hr post-administration of LPS). Results are shown as mean ± SD. *p<0.05, **p<0.01.

The online version of this article includes the following figure supplement(s) for figure 1:

**Figure supplement 1.** Clonogenic and long-term multilineage potential of human and murine hematopoietic stem cells and progenitors (HSC/P) in lymph node (LN).

**Figure supplement 2.** Myeloid progenitor migration to lymph node (LN) in response to lipopolysaccharide (LPS) is independent of NF-κB activation.

the accumulation of a myeloid-committed HSC/P population in human lymphadenitis. Adult inflammatory LN tissues therefore contain an increased number of myeloid-committed HSC/P. This increase can result from either the recruitment of these cells to LN via the bloodstream or the expansion of otherwise rare and already resident myeloid-committed HSC/P in these LN.

The release of HSC/P from BM into the bloodstream follows circadian cycles (*Méndez-Ferrer et al., 2008*) controlled by the activity and fate of inflammatory cells (*Casanova-Acebes et al., 2013*; *Chang et al., 2014*). We postulated that if inflammation is responsible for the recruitment of HSC/P to the LN and possibly other organs, we should be able to recapitulate the process of BM egression, migration, and organ retention in an inflammatory murine model wherein the HSC/P migration process is highly conserved. Since the largest content of HSC/P in human LN was found in biopsies from patients with lymphadenitis, we generated a mouse model of Gram-negative sepsis by injection of *Escherichia coli* LPS into C57Bl/6 mice at the early time point of the circadian HSC/P mobilization cycle (*zeitgeber* time [ZT]) (*Bellet et al., 2013*; *Figure 1B*) and analyzed the rapid migration of HSC/P to different organs. This model has the advantage of requiring one single dose of LPS and provides a relevant approach to the analysis of rapid migration mechanisms deprived of confounding effects of proliferation and/or survival induced by LPS at later time points (*Nagai et al., 2006*; *Zhao et al., 2014*). *E. coli* LPS is able to activate a large number of Toll-like receptors (TLRs), which result in high-level activation of the inflammatory signaling cascade (*Beutler and Rietschel, 2003*). LPS is also a well-known inducer of HSC/P mobilization to PB (*Cline and Golde, 1977*; *Velders et al., 2004*; *Vos et al., 1972*; *Vos and Wilschut, 1979*). In our experiments, the circadian mobilization pattern of HSC/P in the PB was severely modified by the administration of LPS, with the increase in HSC/P appearing later and peaking at ZT10, 6 hr post-administration (*Figure 1C*), coincident with an increased neutrophil count in the PB (*Figure 1D*). We observed similar kinetics of an increased number of HSC/P in the highly vascularized kidney and liver tissues after LPS administration (*Figure 1E, F*), suggesting that the presence of HSC/P in these tissues closely paralleled their presence in the PB. Interestingly, LPS did not elicit a significant change in the level of splenic HSC/P within the first 12 hr after inflammation (*Esplin et al., 2011*; *Wright et al., 2001*; *Figure 1G*).

Notably, when the HSC/P content was reduced in the BM, the kinetics of their subsequent mobilization to the PB was discordant. The BM HSC/P content decreased, which supports the migratory nature of the increased HSC/P in the PB (*Figure 1H*); yet the nadir of the BM HSC/P content occurred as early as 3 hr after LPS administration (at ZT7), returning to normal values by 6 hr (ZT10, *Figure 1H*). The time lapse between the loss of retention of HSC/P in the BM and their presence in the PB circulation suggested that the migration of HSC/P from the BM to the PB required an intermediate step of circulation through other tissues. Based on an earlier description of a lymphatic circulation of HSC/P (*Massberg et al., 2007*), we hypothesized that this delay in the appearance of HSC/P in the PB was due to an intermediate transit of HSC/P through the lymphatic circulation. Indeed, the lymphatic circulation in LPS-treated animals did show a significant increase in the levels of circulating HSC/P in the LN and the thoracic duct compared to controls that closely mirrored the decline of HSC/P in the BM (*Figure 1I, J*).

We next characterized the type of primitive cell populations migrating into the LN via the lymphatic circulation. We first analyzed whether the content of immunophenotypically identifiable BM HSC populations changed concomitantly with the progenitor population changes previously described. LPS induced expansion of BM Lin⁻/c-kit⁺/Sca-1⁺ (LSK) and immunophenotypically identified long-term (LT)-HSC, short-term (ST)-HSC, and multipotential progenitors (MPPs) populations at later time points (Z10–ZT16) (*Figure 1—figure supplement 1E-K*) with no changes in the BM HSC content by ZT7, suggesting a differential effect of LPS signaling on the HSC population. Interestingly, the reduction in the BM content of progenitors was not homogenous throughout the hematopoietic progenitor populations. Confirming the egress of BM CFU-GM described above, the GMP population was significantly decreased by ZT7 (*Figure 1—figure supplement 1F, K*), while the content of immunophenotypically defined common myeloid progenitors (CMPs) only declined by ZT10 (*Figure 1—figure supplement 1F, L*), and the megakaryocyte-erythroid progenitor (MEP) content was increased (*Figure 1—figure supplement 1F, M*), resulting in no significant net changes in the content of BM Lin⁻/c-kit⁺/Sca1⁻ (LK) cells (*Figure 1—figure supplement 1F, N*).

Functional in vivo assays of LN cell suspensions obtained at ZT7 demonstrated that the accumulation of progenitors in LN did not contain any significant numbers of long-term or medium-term repopulating HSC. The analysis of competitive repopulating units (CRUs) in the LN (*Figure 1—figure supplement 2A*) demonstrated that inflamed LN did not contain increased levels of repopulating cells by ZT7 (*Figure 1—figure supplement 2B*). LN contained a transient, ST-myelopoietic progenitor population without medium- or long-term multilineage repopulation ability (*Figure 1—figure supplement 2C*). Lineage analysis of donor-derived circulating cells demonstrated no significant change in T-cell transfer (*Figure 1—figure supplement 2D*) and a diminished transfer of B-cells into the lethally irradiated recipients (*Figure 1*, *Figure 1—figure supplement 2E*), indicating the presence of adoptively transferred lymphoid cells and the absence of mobilization of competent lymphoid progenitors to the inflamed LN. Furthermore, LN SP cells from LPS-treated mice are enriched in LK cells and depleted in LSK cells (*Figure 1—figure supplement 2F*). LN SP cells contain exclusively ST-repopulating progenitors with the ability to differentiate into myeloid cells (data not shown) and are depleted from any significant 8–16 weeks engrafting HSC, as assessed using CRU assays (*Figure 1*, *Figure 1—figure supplement 2F, G*), unlike their BM SP counterparts that are enriched in LSK cells and LT-repopulating activity (*Figure 1*, *Figure 1—figure supplement 2F, H*). These results confirmed that, similar to human inflammatory LN, the LN myeloid progenitors from mice treated with *E. coli* LPS accumulate in inflamed LN and are depleted of stem cell activity. Altogether, these data indicate that LPS induces a selective lymphatic circulation of myeloid committed progenitors, but not other types of HSC/P populations.

To explore the nature of the circuit of the myeloid progenitor migration to LN, we first analyzed the ability of HSC/P to seed LN in non-myeloablated mice. For this experiment, we labeled C57Bl/6-BM-derived lineage negative (Lin⁻) cells containing the HSC/P fraction with the lipophilic dye 1,1'-dioctadecyl-3,3,3',3'-tetramethylindocarbocyanine perchlorate (DiI) and adoptively transferred them into unmanipulated lymphatic vessel reporter *Lyve1ᵉᴳᶠᴾ* mice. *Lyve1ᵉᴳᶠᴾ* knock-in mice display enhanced green fluorescent protein (eGFP) fluorescence driven by the promoter/enhancer of the lymphatic vessel hyaluronan receptor 1 Lyve-1 identifying lymphatic endothelial cells (*Pham et al., 2010*). We analyzed the 17 hr homing of Lin⁻ cells to the BM and LN (*Figure 2A*). The homing of cells to the BM in mice treated with LPS was reduced by ~65% compared with their vehicle-treated counterparts (*Figure 2B*). To determine whether Lin⁻/DiI⁺ homed cells leave BM in response to LPS, we first determined the existence of transcortical vessels (*Figure 2C*; *Grüneboom et al., 2019*) and the presence of lymphatic vasculature inside the bone by two-photon microscopy in mice either treated with PBS or LPS (*Figure 2D, E* and *Videos 1* and *2*). *Lyve1ᵉᴳᶠᴾ* knock-in mice revealed that the bone of LPS-treated animals contains a Lyve1+network, which was only rarely identified in PBS-treated mice (*Figure 2D*), suggesting that LPS-induced inflammation may upregulate the expression of the hyaluronan receptor Lyve1 and render patent a pre-existing network of Lyve1+bone cells. Interestingly, Lin⁻/DiI⁺ homed cells were located closer to the endosteum in response to LPS at as early as 1.5 hr after administration of LPS (*Figure 2E*). Quantification of the distance of Lin⁻/DiI⁺ homed cells to endosteum area showed significant differences between PBS and LPS treatment, indicating increased proximity to the endosteum area after LPS (*Figure 2F*). We found, albeit at a very low frequency, tiny lymphatics scattered and projected inside the bone (*Figure 2G* and *Video 3*). On the other hand, the seeding of BM-derived Lin⁻/DiI⁺ cells into LN increased approximately threefold,

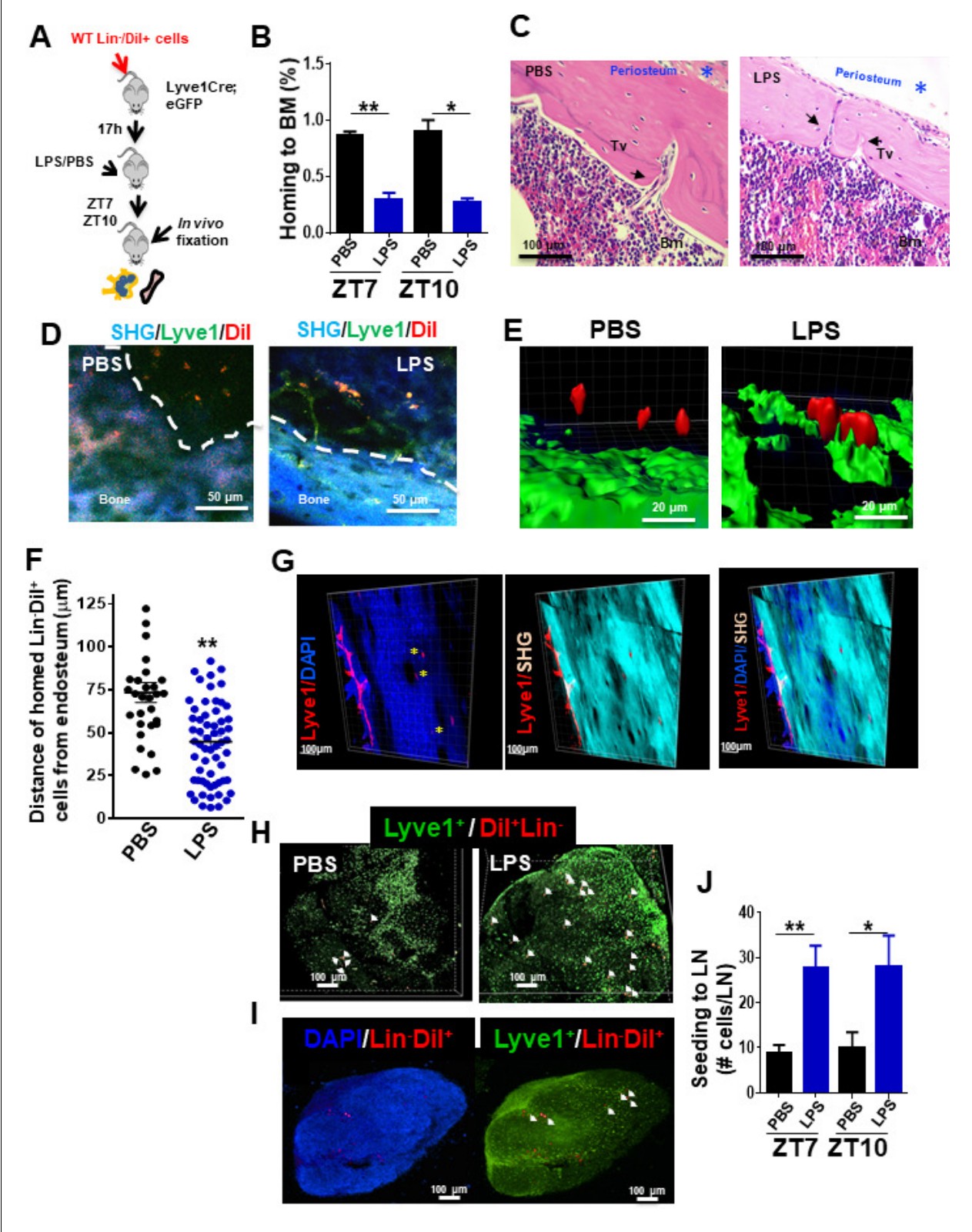

**Figure 2.** Draining of bone marrow (BM)-derived lineage negative cells into lymphatics. (**A**) Schema for adoptive transfer of BM-derived lineage negative cells (Lin⁻) labeled with 1,1'-dioctadecyl-3,3,3',3'-tetramethylindocarbocyanine perchlorate (Dil) dye into lymphatic endothelium reporter *Lyve1ᵉᴳᶠᴾ* mice. By 16 hr after cell transplantation, lipopolysaccharide (LPS) or PBS were administered into the *Lyve1ᵉᴳᶠᴾ* mice at zeitgeber time (ZT)4 (10 am; time of LPS administration). BM and lymph node (LN) cells were analyzed at ZT5.5 (11.30 am, 1.5 hr after LPS administration), ZT7 (1 pm, 3 hr

*Figure 2 continued on next page*

Figure 2 continued

after LPS administration), and ZT10 (4 pm, 6 hr after LPS administration) for labeled Lin⁻ cells (Lin⁻/DiI⁺). (B) Frequency of Lin⁻/DiI⁺ homed to BM (solid bars) (mosaic bars) after PBS (black solid bar) or LPS (blue solid bar) administration at ZT7 (1 pm, 3 hr after LPS administration) and ZT10 (4 pm, 6 hr after LPS administration). (C) Representative 3D reconstruction images of the whole bone by two-photon microscopy showing lymphatic vessels (Lyve1) surface marker (red), nuclei with DAPI (blue), and cortical bone with second harmonic signal (SHG, light blue). The scale bars, 100 μm. (D, i–iv) Intravital two-photon microscopy imaging (IVM) of long bones from *Lyve1^eGFP* mice showing lymphatic vessels (displayed in green) near to the surface of the bone (blue, detected by SHG signal) and homed Lin⁻/DiI⁺ cells (displayed in red). (i, iii) IVM of PBS specimen. (ii, iv) IVM of LPS specimen. (E, F) Analysis and quantification of the distance of homed Lin⁻/DiI⁺ (red) cells to *Lyve1^eGFP* (green) cells after PBS/LPS administration at ZT5.5 (11.30 am, 1.5 hr after LPS administration) analyzed by Imaris 7.7.2 software. (G) Two-photon microscopy examples of images of longitudinal femoral sections stained with anti-Lyve1 antibody and DAPI, and analyzed for specific fluorescence signal and SHG for cortical bone. (H, I) Representative of 3D reconstitution images of PBS- and LPS-treated LN tissues (H) and cross-sections of LPS-treated LN (I) analyzed by confocal microscopy showing the location of mobilized Lin⁻/DiI⁺ cells (red; nucleus stained by DAPI in blue) in relation with Lyve1+ cells (green; nucleus stained by DAPI in blue). The Z-stack dimensions of upper panels were X = 1266.95 μm, Y = 1266.95 μm, and Z = 344 μm. Calibrate: XY = 2.47 μm and Z = 4 μm. Resolution: 512 × 512 × 86. The Z-stack dimensions of lower panels were: X = 1259.36 μm, Y = 1259.36 μm, and Z = 132 μm. Calibrate: XY = 2.46 μm and Z = 4 μm. Resolution of images was 512 × 512 × 86. (J) Absolute count of mobilized Lin⁻/DiI⁺ cells counted within LN at ZT7 (1 pm, 3 hr after LPS administration, solid bars) and ZT10 (4 pm, 6 hr after LPS administration, mosaic bars) after PBS/LPS administration. N = 4–14 LNs analyzed per time point in a minimum of three mice per group and/or time point. Graph data depict mean ± SD. *p<0.05, **p<0.01.

which mirrored the decline in BM homing (*Figure 2H–J*). Histological analysis of BM-derived Lin⁻/DiI⁺ cells within the LN by confocal microscopy showed that the migrated HSP/C are spatially positioned in the cortex area surrounding primary follicles (*Figure 2I*), consistent with localization in T-cell zone for antigen presentation. These findings strongly suggest that the rapid egress of hematopoietic progenitors from BM during inflammation may indeed occur through bone lymphatics draining into LN.

## Systemic inflammation recruits dendritic cell-committed phenotypic progenitors to LN

To determine the potential of the myeloid progenitors mobilized to the LN, we further determined their in vitro and in vivo differentiation profile. To this end, we analyzed the differentiation capabilities of LN myeloid progenitors in specific cytokine-driven clonal assays in methylcellulose assays. The majority of the differentially accumulated myeloid progenitors in LN by 3 hr post-administration of LPS were granulocyte-macrophage progenitors (CFU-GM) and in a much lesser degree unipotent granulocyte progenitors (GPs, CFU-G) with no differential accumulation of unipotent macrophage progenitors (MPs, CFU-M) (*Figure 3A*). Next, we investigated whether GMP were able to home and migrate to LN after in vivo administration of LPS. For this purpose, we adoptively transferred sorted β-actin/eGFP transgenic GMPs into congenic mice. Transgenic GMPs were allowed to home to the BM and after 17 hr recipient mice were treated with a single dose of LPS or vehicle control. On day 7 after PBS or LPS administration, murine BM and LN were analyzed for donor-derived granulocytes (Gr1⁺⁺/CD11b⁺/CD11c^neg), macrophages (Gr1^dim/CD11b⁺/CD11c^neg), and lymphoid tissue cDC (Gr1^neg/CD11b⁺/CD11c⁺) by flow cytometry (*Figure 3B*). We found that LPS induced differential donor-derived specific GMP differentiation toward the formation and retention of cDC in LN (*Figure 3C*), but not in the BM (*Figure 3D*). The content of macrophages and granulocytes did not significantly change with LPS in either LN or BM (*Figure 3C, D*), confirming the specific nature of the lymphoid-tissue cDC differentiation of mobilized GMP in LPS-treated mice.

To elucidate whether the LN cDC content was dependent on migration of committed cDC

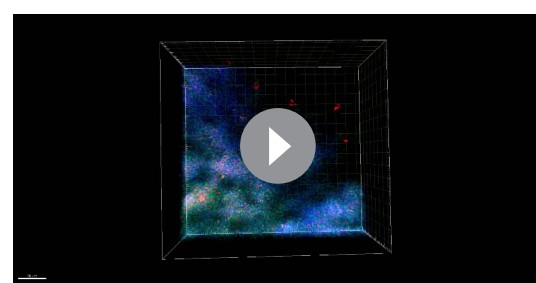

**Video 1.** Representative example of multiphoton microscopy processed with Imaris software of a femur from a *Lyve1^eGFP* mouse treated with PBS at 1 hr after administration (11 am, 1 hr after lipopolysaccharide administration). Bone tissue is identified as second-harmonic signal (blue). Imaris software was used to measure distance between DiI-labeled cells and green fluorescent protein-positive lymphatic vessels using 3D images.

https://elifesciences.org/articles/66190#video1

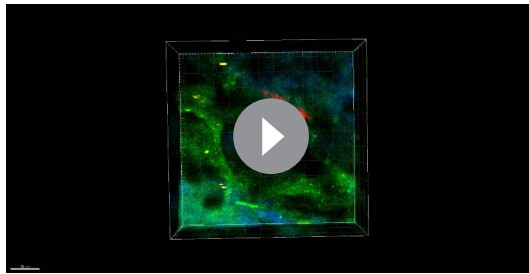

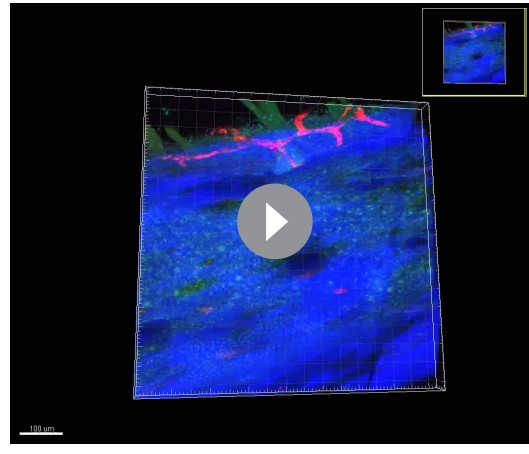

**Video 2.** Representative example of multiphoton microscopy processed with Imaris software of a femur from a *Lyve1^eGFP* mouse treated with LPS at 1 hr after administration (zeitgeber time 5). Bone tissue is identified as second-harmonic signal (blue). Imaris software was used to measure distance between DiI-labeled cells and green fluorescent protein-positive lymphatic vessels using 3D images.

https://elifesciences.org/articles/66190#video2

**Video 3.** Representative example of transcortical lymphatics discovered by staining with anti-Lyve1 (red) in wild-type C57Bl/6 mice. Bone tissue is identified as second-harmonic signal (blue). Green signal is autofluorescence.

https://elifesciences.org/articles/66190#video3

precursors opposed to local specification of migrated macrophages or MPs, we analyzed the migration of macrophage-dendritic progenitors (MDPs). BM MDPs represent a progenitor population that can differentiate into monocytes/macrophages or directly into cDCs without intermediate macrophage specification (*Fogg et al., 2006*; *Geissmann et al., 2003*; *Waskow et al., 2008*). MDPs are characterized by high expression of the chemokine receptor Cx3cr1 (in *CX₃CR-1^GFP* reporter mice), c-fms (CD115) and Flt3, and intermediate expression of c-Kit. Serial gating of lineage⁻/Cx3cr1-GFP⁺⁺/c-Kit^int cells (*Figure 3E*, P2) showed an approximately threefold accumulation of CD115⁺⁺/Flt3⁺⁺ cells in LN and a concomitant 65% depletion in the BM as early as 3 hr after LPS administration (*Figure 3F, G*, *Figure 3—figure supplement 1A, B*). In the absence of significant changes in the LN content of macrophages, these data demonstrate that LPS-induced systemic inflammation results in robust and specific recruitment of phenotypic GMP that are BM cDC-committed progenitors to the LN.

## Myeloid progenitor migration to LN is Traf6-dependent and NF-κB-independent

Immune cells and HSC/P express TLR (*Beutler and Rietschel, 2003*; *Nagai et al., 2006*; *Takeuchi and Akira, 2010*), which act as microorganism sensors. LPS stimulation of TLR recruits MyD88 and TRIF through the canonical and endosomal pathways, respectively. Both adaptors subsequently recruit *TRAF6*, which acts as the molecular hub of both signaling branches (*Akira et al., 2001*; *Kawai and Akira, 2006*). To determine whether Traf6 deficiency might affect the migration of HSC/P in response to LPS, we exploited an animal model in which Traf6 is deleted only in hematopoietic cells (*Kobayashi et al., 2003*; *Figure 4A*, *Figure 4—figure supplement 1A*). LN from Mx1Cre;WT chimeric mice after LPS administration (at ZT7, corresponding with the peak of progenitor content in LN, *Figure 1H*) revealed a threefold increase in the frequency of CFU-GM. This increase was completely abrogated by the deficiency of Traf6 in hematopoietic cells (Mx1Cre;*Traf6^Δ/Δ* animals, *Figure 4B*), indicating that the signals that result in CFU-GM mobilization to LN are mediated by hematopoietic Traf6.

Although HSC/P respond directly to PAMPs such as LPS (*Nagai et al., 2006*; *Zhao et al., 2014*), direct Gram-negative infection-derived LPS sensing by HSC/P does not play an essential role in emergency granulopoiesis, but rather requires TLR4-dependent signals within the microenvironment (*Boettcher et al., 2012*; *Kwak et al., 2015*). To further test whether the hematopoietic Traf6-dependent response to LPS resulting in mobilization of GMP from the BM to the LN by ZT7 is indeed cell autonomous and not determined by the microenvironment, we used the conditional Traf6-deficiency model and analyzed the in vitro migration of myeloid progenitors toward chemoattractant gradients

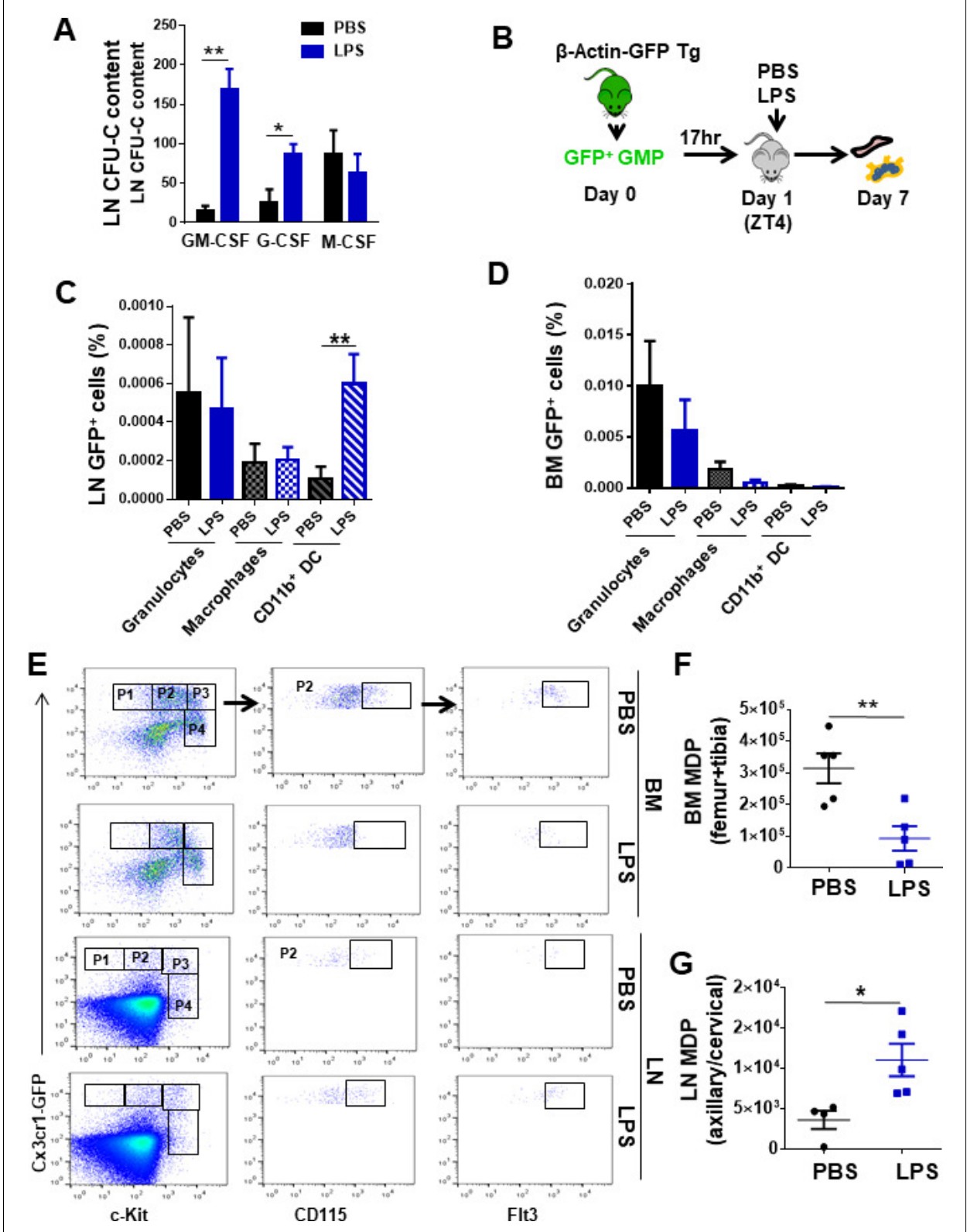

**Figure 3.** Lymph node (LN)-mobilized granulocyte-macrophage progenitors (GMPs) preferentially differentiate into dendritic cells. (**A**) Comparative quantification of the content of bipotent and unipotent myeloid progenitors in LN at zeitgeber time (ZT)7 (1 pm, 3 hr after lipopolysaccharide (LPS) administration) after PBS/LPS administration. (**B**) Schema of isolation and transfer of bone marrow (BM)-derived GMP from β-actin-GFP reporter mice (CD45.2[+]) into C57Bl/6 (CD45.2[+]) mice (2 × 10[5] GFP[+]-GMP cells/mouse, n = 3 mice per group). After BM homing (17 hr), mice were treated with single

*Figure 3 continued on next page*

*Figure 3 continued*

and low dose of LPS (5 mg/kg) at ZT4 (day 1) and 7 days later BM and LN tissues were analyzed for green fluorescent protein (GFP) expression in myeloid populations by flow cytometry. (C, D) Graphs represent the percentage of GFP$^+$-GMP differentiated to granulocytes (solid bars, Gr1$^{++}$CD11b$^+$CD11c$^-$), macrophages (left mosaic bars, Gr1$^{dim}$CD11b$^+$CD11c$^{neg}$), and cDC (right mosaic bars, Gr1$^-$CD11b$^+$ CD11c$^+$) 7 days post-transferring after PBS/LPS administration by flow cytometry into LN (C) and BM (D). (E) Fluorescence-activated cell sorter strategy for macrophage-dendritic progenitor (MDP) content in BM and LN tissues from $CX_3CR$-$1^{GFP}$ reporter mice (4–5 mice per group). Phenotypically, MDPs are defined as lineage-negative with high expression of the chemokine receptor Cx3cr1, c-fms (CD115) and Flt3 (P2), and intermediate expression of c-Kit. (F, G) Graphs show absolute numbers of MDP present in BM (F) and LN (G) 3 hr later (ZT7 [3 hr]) after PBS (black circles) or LPS (blue squares) administration. Values represent mean ± SD. *p<0.05, **p<0.01.

The online version of this article includes the following figure supplement(s) for figure 3:

**Figure supplement 1.** Lipopolysaccharide (LPS) induces rapid migration of macrophage-dendritic progenitors (MDPs) to regional lymph node (LN) chains.

---

generated by LPS-stimulated BM or LN cells in assays designed to identify the hematopoietic cell population affected by LPS (*Figure 4C, E, G*). We found that the cell-autonomous deficiency of Traf6 resulted in a relative decrease in migration of ~40% (*Figure 4D*) of BM myeloid progenitors in the presence of LPS, indicating that *Traf6* is required for LPS-dependent cell-autonomous BM myeloid progenitor migration. Interestingly, analysis of non-cell-autonomous migration of BM myeloid progenitors demonstrated that LN-derived cells generated more potent chemoattractant signals, resulting in a much larger migration of WT BM myeloid progenitors (fivefold higher, ~30%) during the same period (*Figure 4E, F*), which was drastically diminished (~50% reduction) by Traf6-deficiency in LN cells, but not when using control BM cells as chemoattractant source (*Figure 4E, F*). These data indicate that although LPS-mobilized myeloid progenitors depend on both cell-autonomous and non-cell-autonomous Traf6-dependent signals, the chemoattractant gradient generated by LPS on LN cells is the predominant effect responsible for Traf6-dependent myeloid progenitor migration. Interestingly, the non-cell-autonomous effect does not seem to be due to secretion of the chemokine inducer IFN-γ, which was not detectable in LN-derived supernatant (*Figure 4—figure supplement 1B*), strongly suggesting that activated LN NK cells may not be responsible for the migration of BM myeloid progenitors. And, similarly, the effect of Traf6 expression on non-cell-autonomous, LN-mediated migration did not depend on the secretion of IL-1α, IL-2, IL-13, IL-4, TNF-α, or IL-10 regulatory cytokines of inflammatory processes (*Figure 4—figure supplement 1C–H*).

To delineate the resident LN cell population responsible for the migration of GMPs into the LN, we isolated putative effector cell populations representing 1% or more of the cellularity of either BM or LN tissues. We isolated T-cells (CD3e$^+$), B-cells (B220$^+$), and myeloid cells (CD11b$^+$) from LN of WT or Traf6$^{\Delta/\Delta}$ mice and layered input cell equivalents on the bottom of the chamber with LPS, as in the previous experiments (*Figure 4G*). Although only ~1% of LN cells are myeloid, we observed that LN CD11b$^+$ cells, but not B- or T-cells, from Traf6$^{\Delta/\Delta}$ mice can recapitulate the same reduction of progenitor migration achieved by complete LN tissue (*Figure 4H*). To confirm that a Traf6-dependent signaling in LN CD11b + cells is responsible for myeloid progenitor mobilization and eliminate the possible inflammatory effect of previous treatment with polyI:C in Mx1-Cre transgenic mice, we crossed *Traf6$^{flox/flox}$* mice with *Lyz2$^{Cre}$* transgenic mice (*Clausen et al., 1999*; *Cross et al., 1988*) and analyzed the migration to LN after LPS administration in mature myeloid lineage-specific Traf6-deficient (*Lyz2$^{Cre}$;Traf6$^{flox/flox}$*) mice. Mature myeloid lineage-specific deletion of *Traf6* abrogated the migration of myeloid progenitors to LN in response to LPS (*Figure 4I*). A major consequence of the deficiency of Traf6 in mature myeloid lineage-specific cells was an increase in the endotoxemia-dependent mortality (*Figure 4J*), indicating that Traf6 expression in mature myeloid cells is required for both migration of myeloid progenitors to LN and protection of LPS-induced mortality. Altogether, these data indicate that LPS/Traf6 signaling is required for migration of myeloid progenitors through predominantly long-range acting, mature myeloid lineage-dependent chemoattractant signals, and that LPS/Traf6 signaling in Lyz2-expressing cells is protective against endotoxin-induced inflammation.

Activation of TLRs conserves inflammatory pathways that culminate in the activation of the NF-κB transcription factors (*Karin and Greten, 2005*). The LPS binds TLR4/MD2 complexes on the cell surface, and through a series of adaptors and kinases recruits Traf6. By an E3 ligase-dependent

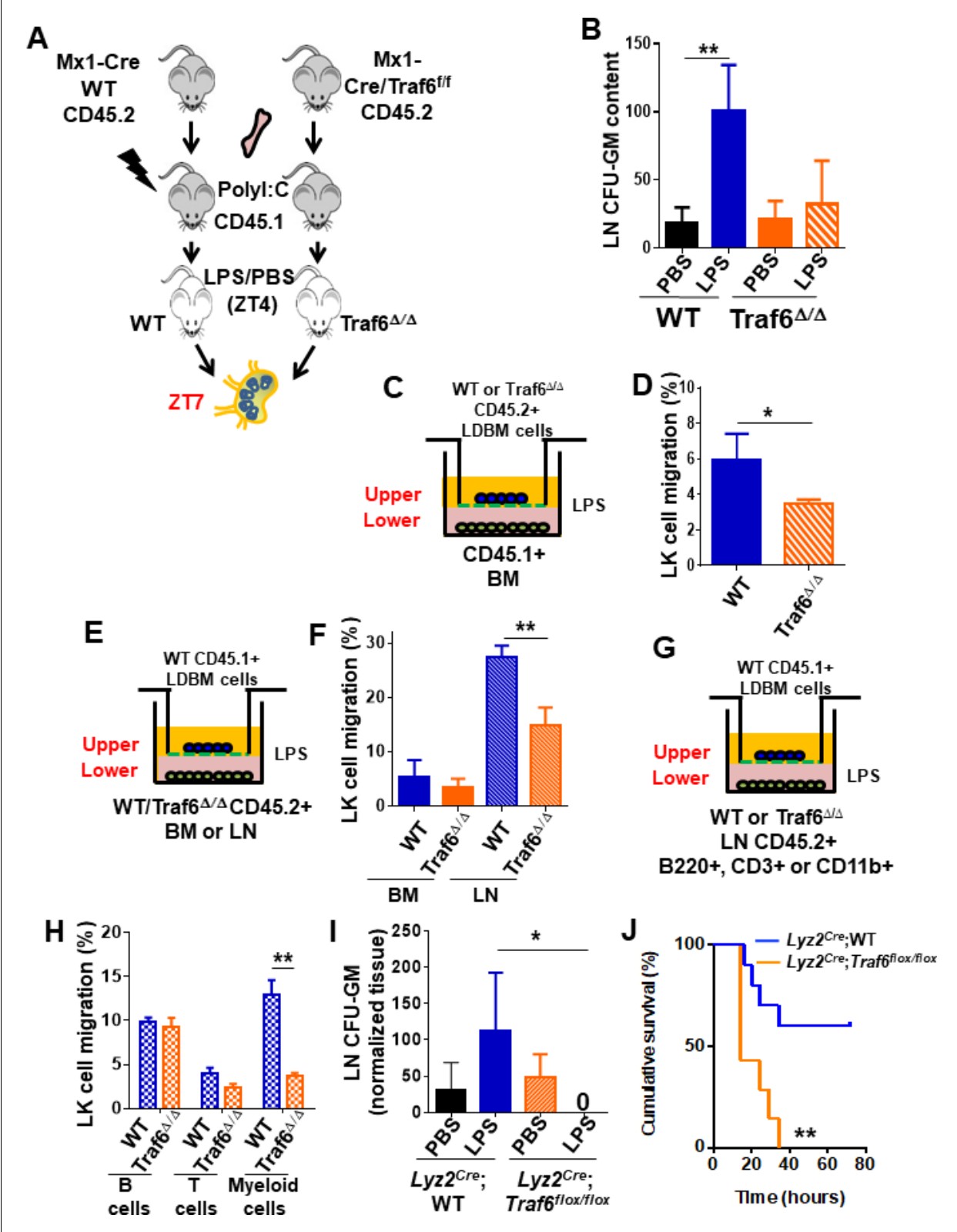

**Figure 4.** Traf6 is a key regulator for migration of bone marrow (BM)-derived myeloid progenitors to lymph nodes (LNs) in a non-cell-autonomous manner. (**A**) Schema of full chimeric mice made by non-competitive transplantation of CD45.2⁺ *Mx1Cre;WT* and *Mx1Cre;Traf6^flox/flox* BM cells into lethally irradiated CD45.1⁺ B6.SJL^Ptprca Pep3b/BoyJ. Six weeks later Traf6 gene were deleted by intraperitoneal injection of poly(I:C). 1 week later we performed PBS/lipopolysaccharide (LPS) injection early in the rest phase (zeitgeber time [ZT]4 [10 am, time of LPS administration]) and LN-contained

*Figure 4 continued on next page*

*Figure 4 continued*

myeloid progenitors at ZT7 (1 pm, 3 hr after LPS administration) was scored by colony-forming unit (CFU) assay. (B) Absolute number of CFU-GM present in LN from wild-type (WT) (solid bars) and Traf6$^{\Delta/\Delta}$ (orange bars) full chimeric mice (n = 6–7 mice per group) after PBS (black and orange solid bars) or LPS (blue and mosaic bars) administration. (C–H) In vitro transwell migration assay for BM-derived LK cells. (C) Experimental design for migration of WT or Traf6$^{\Delta/\Delta}$ low-density (LD) BM cells (CD45.2$^+$) toward a WT microenvironment generated by BM (CD45.1$^+$) in the presence of LPS for 4 hr. (D) Graph represents the percentage migrated LK from WT (blue solid bar) or Traf6$^{\Delta/\Delta}$ (orange mosaic bar) low-density BM (LDBM) cells to the bottom as depicted in (C). (E) Experimental design for migration of WT LDBM cells (CD45.1$^+$) toward a gradient generated by WT or Traf6$^{\Delta/\Delta}$ (CD45.2$^+$) BM or LN cells in the presence of LPS for 4 hr. (F) Graph represents the percentage of LDBM LK migrated to the BM bottom (solid bars) or LN bottom (mosaic bars) as schemed in (E). (G) Experimental design for WT LDBM cells (CD45.1$^+$) migration toward gradient generated by WT or Traf6$^{\Delta/\Delta}$ LN-derived T-cells (CD45.2$^+$/CD3e$^+$/CD11b$^-$/B220$^-$) or B-cells (CD45.2$^+$/CD3e$^-$CD11b$^-$/B220$^+$) or myeloid cells (CD45.2$^+$/CD3e$^-$/CD11b$^+$/B220$^-$) in the presence of LPS for 4 hr. (H) Graph represents the percentage of migrated LK LDBM to the WT LN bottom (blue mosaic bars) or Traf6$^{\Delta/\Delta}$ LN bottom (orange mosaic bars) as schemed in (G). In all cases, LK cell migration was determined by CD45 allotype analysis using flow cytometry in triplicate. (I) Absolute number of CFU-GM present in LN from Lyz2$^{Cre}$;WT (solid bars) and Lyz2$^{Cre}$;Traf6$^{flox/flox}$ (mosaic bars) full chimeric mice after PBS/LPS administration at ZT7 (3 hr). (J) Graph represents cumulative survival of Lyz2$^{Cre}$;WT (blue line) and Lyz2$^{Cre}$;Traf6$^{flox/flox}$ (orange line) after 10 mg/kg of LPS. (J) Survival curve after 30 mg/kg of b.w. injection in Lyz2$^{Cre}$;WT (blue line) or Lyz2$^{Cre}$;Traf6$^{flox/flox}$ (orange line). Values are shown as mean ± SD of two independent experiments with a minimum of three mice or replicates per group and experiment. *p<0.05, **p<0.01.

The online version of this article includes the following figure supplement(s) for figure 4:

**Figure supplement 1.** Myeloid progenitor migration to lymph node (LN) in response to lipopolysaccharide (LPS) is independent of NF-κB activation.

**Figure supplement 2.** Inflammation induces temporal changes in chemokine and cytokine signatures in bone marrow (BM) and lymph node (LN).

mechanism, Traf6 activates the IκB kinase (IKK) complex, which initiates IκBα degradation, and our group has demonstrated that its constitutive deficiency results in reduced HSC quiescence and increased progenitor proliferation (*Fang et al., 2018*). Subsequent nuclear translocation of NF-κB transcription factors results in the expression of cytokine and chemokine genes. To determine whether emergent NF-κB signaling is responsible downstream of LPS/*Traf6* for the LPS-induced LN migratory effect of myeloid progenitors, we overexpressed a degradation-resistant mutant of IκBα (IκBα super-repressor [IκBα$_{SR}$]), in primary murine progenitors, which were then differentiated into macrophages/cDC by macrophage colony-stimulating factor (M-CSF) (*O'Keeffe et al., 2010*; *Figure 4—figure supplement 1I*). Analysis of LPS-driven migration in vitro (*Figure 4—figure supplement 1J*) demonstrated that the expression of IκBα$_{SR}$ does not reduce the effect of LPS on the migration of myeloid progenitors toward LPS-stimulated macrophages/cDC (*Figure 4—figure supplement 1K, L*), indicating that NF-κB transcription factors are dispensable for myeloid progenitor migration. In contrast, inhibition of intracellular protein traffic using monensin dramatically decreased myeloid precursor migration (*Figure 4—figure supplement 1K, L*), suggesting that intracellular protein trafficking is necessary for the migration phenotype. Collectively, these data indicate that LPS-induced myeloid progenitor migration occurs through an NF-κB-independent, intracellular protein traffic-dependent pathway, and suggests that the progenitor-mobilizing effect of LPS may not require transcriptional activation, depending rather on the intracellular traffic of secreted proteins.

## Myeloid progenitors home into LN in a Ccl19/Ccr7-dependent fashion but independently of L-selectin

The secretome of myeloid cells includes multiple cytokines/chemokines with short- and long-range activities on activation, proliferation, survival, differentiation, and migration of target cells. Specifically, secreted chemokines stimulate migration of target cells following chemokines to the areas of highest concentration. It has been described that hematopoietic progenitor migration is dependent on Cxcl12 gradients (*Greenbaum et al., 2013*; *Méndez-Ferrer et al., 2010*). However, by ZT7, LPS induced upregulation of Cxcl12 expression in BM, but not in LN, indicating that Cxcl12 tissue concentrations per se could not explain the mechanism of migration to LN (*Figure 4—figure supplement 2A*). An array of tests on secreted chemokines and cytokines demonstrated distinct secretome signatures between BM and LN tissues after LPS administration (*Figure 4—figure supplement 2B–N*). As expected, LPS induced upregulation after administration of several myeloid cell cytokines and chemokines with ability to recruit and differentiate macrophages and cDC in the extracellular fluid of LN rather than BM as early as 1 hr after LPS challenge (*Figure 4—figure supplement 2C–J*). However, none of these candidate cytokines/chemokines were found to consistently generate a differential tissue concentration in vivo between LN and BM at both ZT5 and ZT7 (*Figure 4—figure*

*supplement 2B*). Similar to Cxcl12, some cytokines/chemokines with potential chemoattractant ability were also found to be upregulated in BM rather than in LN or in both tissues similarly (*Figure 4—figure supplement 2K–N*). The lack of in vivo tissue differential levels strongly suggested that these BM-derived cytokines or chemokines, although likely to play a role in the LPS-mediated inflammatory response, were unlikely to be responsible for the attraction of BM myeloid progenitors to the LN.

The C-C chemokine receptor type 7 (Ccr7) ligand macrophage-inflammatory protein (MIP)-3b/Ccl19 has been reported as a chemoattractant for BM and cord blood $CD34^+$ cells in vitro, mainly CFU-GM (*Kim et al., 1998*), and to direct DCs to LNs and elicit an adaptive immune response (*Förster et al., 1999*). Analysis of Ccl19 in the extracellular fluid of the femoral cavity, LN, and plasma demonstrated that in vivo administration of LPS promotes a secretion of Ccl19 in LN when compared with BM and PB (*Figure 5A*). This differential secretion is specific to Ccl19 since Ccl21, a highly related chemokine, did not show the formation of similar differential tissue concentrations in LN after LPS administration (*Figure 5—figure supplement 1A*). Ccl19 is secreted by LN myeloid cells after LPS stimulation and depends on Traf6 expression (*Figure 5—figure supplement 1B*). Ccr7-mediated signals control the migration of immune cells to secondary lymphoid organs such as LN, facilitating efficient surveillance and targeted cellular response (*Förster et al., 2008*). Also, LPS upregulates membrane Ccr7 expression on cDC and their committed progenitors (*Schmid et al., 2011*). We therefore hypothesized that LN trafficking of phenotypic GMP/MDP is regulated by Ccr7, and that therefore the Ccl19/Ccr7 axis might explain the coexistence of cell-autonomous and non-cell-autonomous mechanisms required for GMP migration from the BM to LN in response to LPS. To test our hypothesis, we first analyzed whether the specific deficiency of either Ccl19 or Ccr7 modified the level of progenitor migration to LN. To prevent the interference of long-term deficiencies of Ccl19 and Ccr7 expression described in deficient murine models (*Förster et al., 1999*; *Mori et al., 2001*), we performed short-term in vivo neutralization of Ccl19 ligand or the Ccr7 receptor by using specific antibodies or isotype controls (*Figure 5B*) and determined the content of CFU-GM in LN after LPS challenge or PBS control. We administered an anti-Ccl19 and an anti-Ccr7 neutralizing antibody (or their controls) twice within 15 hr before LPS administration. We found a dramatic reduction (>90%) in the number of CFU-GM in the LN of LPS-, anti-Ccl19-treated animals by ZT7 (*Figure 5C*). Also, we confirmed that Ccr7-expressing GMP in BM rapidly decreased in response to LPS and increased in regional draining LN chains (*Figure 5—figure supplement 1C–E*). The abrogation of accumulation of progenitors in LN was reproduced by the administration of anti-Ccr7 (*Figure 5D*). Second, we analyzed the membrane expression of Ccr7 on BM-derived GMP, CMP, and MEP from Mx1Cre;WT or Mx1Cre;*Traf6*$^{\Delta/\Delta}$ mice, with or without LPS stimulation. Membrane Ccr7 levels were significantly upregulated as early as 1 hr after LPS administration on GMP in LPS-treated WT mice. Such upregulation was abrogated in LPS-treated Traf6-deficient GMP (*Figure 5—figure supplement 1F-G*). Finally, we confirmed that hematopoietic chimeric Ccl19$^{-/-}$ animals did not mount a migratory response of myeloid progenitors from BM to LN in response to LPS (*Figure 5E, F*). Together, these data indicate that the rapid migration of BM myeloid progenitors to LN depends on the expression of the chemokine Ccl19 and its receptor Ccr7.

Traffic of myeloid progenitors to regional LNs was recapitulated in mice receiving intrafemoral adoptive transfer of GMP (*Figure 5G, H*). In these mice, in vivo L-selectin blockade did not abrogate GMP migration to regional LN while sinusoidal-dependent B-lymphocyte mobilization into regional LN was significantly impaired (*Figure 5H, I*), indicating that the migration of BM myeloid progenitors, unlike B-cells, into the regional lymphatic circulation is L-selectin independent and therefore unlikely to be mediated by LN high endothelial venules (*Rosen, 2004*). Altogether, these data strongly indicated that Ccl19/Ccr7 chemokine signaling is required for the rapid migration of myeloid progenitors to LN upon LPS administration. Given the strong time association of these events, these data support a role for the Ccr7-dependent early traffic of myeloid progenitors in the amelioration or delay of the endotoxic shock induced by LPS.

## Ccl19 is expressed and pre-stored in cDC2 and released upon activation of IKK/SNAP23

Chemokine secretion requires endosomal fusion with the membrane, which can be detected by exposure of the phosphatidylserine (PS)-rich inner leaflet of the endosomes to the external surface of the cell membrane, providing a venue to determine what cell types were responsible for the secretion of Ccl19. We found an increase in the levels of PS residues on the outer membrane leaflet

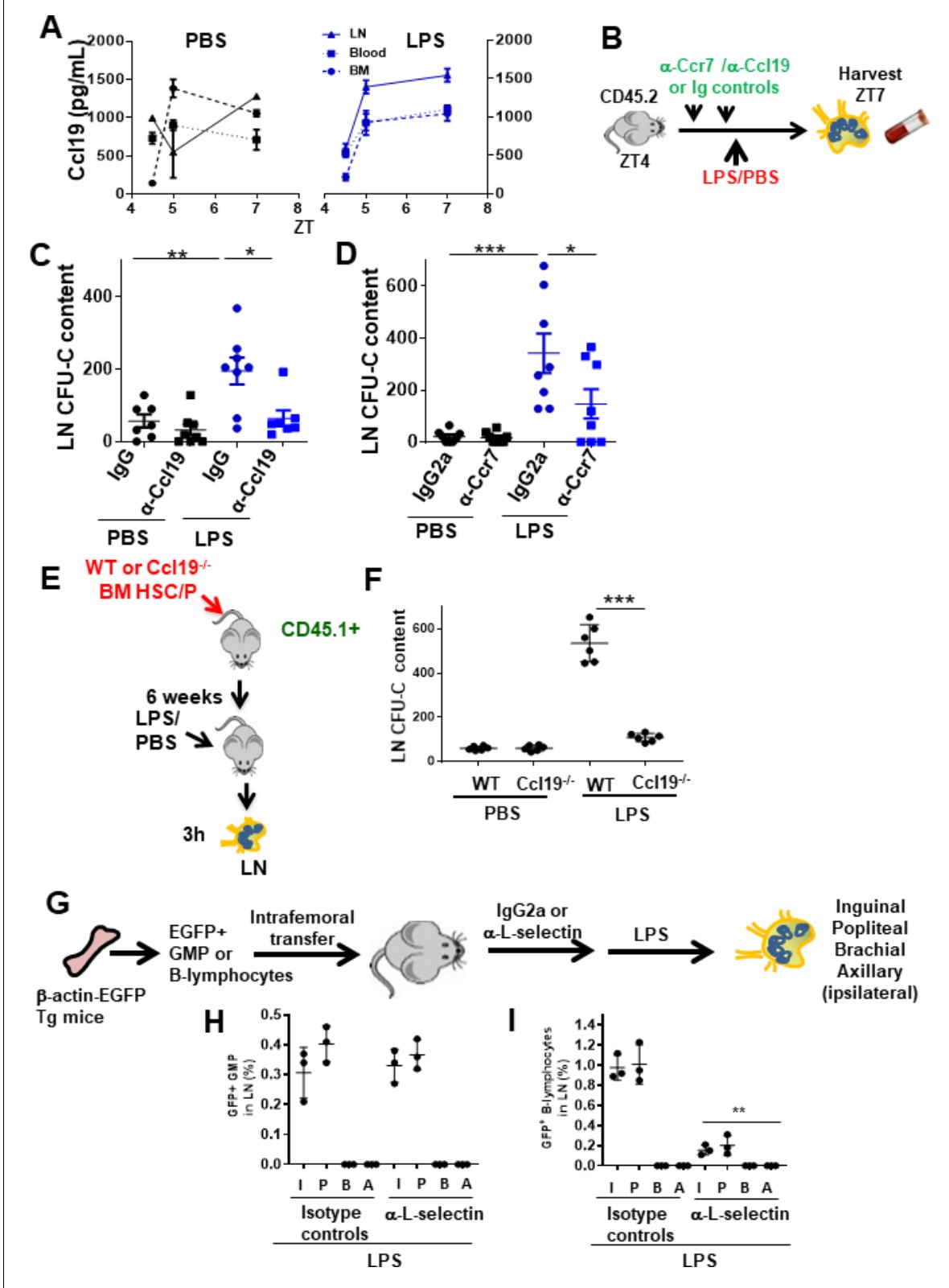

**Figure 5.** Granulocyte-macrophage progenitor (GMP) cells drain into local lymphatics and not blood circulation in early inflammation via Ccl19/Ccr7. (A) Graph represents soluble Ccl19 chemokine in femoral or lymph node (LN) extracellular fluid and peripheral blood (PB) plasma after PBS (black lines) or lipopolysaccharide (LPS) (blue lines) administration at different circadian cycle times (zeitgeber time [ZT]4.5 [10.30 am, 0.5 hr after LPS administration], ZT5 [11 am, 1 hr after LPS administration], and ZT7 [1 pm, 3 hr after LPS administration]). (B) Strategy for in vivo neutralization of Ccr7 receptor or Ccl19

*Figure 5 continued on next page*

**Figure 5 continued**

ligand by injections of anti-Ccr7 antibody or anti-Ccl19 antibody (50 μg/dose, two doses) into C57Bl/6 mice. One day after the last dose of antibodies, PBS or LPS was administered at ZT4 (10 am, time of LPS administration), and the myeloid progenitors-circulating cells from the LN and PB were measured by colony-forming unit (CFU) assay at ZT7 (1 pm, 3 hr after LPS administration). (C, D) Absolute number of progenitors present into LN from neutralized mice with anti-Ccl19/IgG (left graph) or anti-Ccr7/IgG2a (right graph) after PBS (black) or LPS (blue) administration as depicted in (B). (E) Generation of hematopoietic chimeric Ccl19 expressing (wild-type [WT]) or not (Ccl19[-/-]) mice and isolation of LNs after administration of PBS or LPS. (F) CFU content of LN from either WT or Ccl19[-/-] hematopoietic chimeric animals treated with PBS or LPS. (G) Experimental design to analyze L-selectin dependence of femoral GMP migration to regional (or distant) LN after LPS administration. (H, I) Percentages of GFP[+] cells in LN after administration of an isotype control or anti-L-selectin antibodies. (H) Frequency of GFP[+]GMP cells in regional LN after administration of LPS was not modified by L-selectin blockade in vivo. (I) Inhibition of the migration of GFP[+] B-lymphocytes to regional LNs in mice pre-treated with anti-L-selectin antibody. In (B) and (C), LN were collected at ZT7 or 3 hr after LPS. Values represent mean ± SD of replicates in two or three independent experiments. *p<0.05, **p<0.01, ***p<0.001.

The online version of this article includes the following figure supplement(s) for figure 5:

**Figure supplement 1.** Myeloid expression of Ccl19 ligand in lymph node (LN) and short-term differentiation of pre-treated granulocyte-macrophage progenitors (GMPs).

---

of WT LN cDC2 (defined as CD11b[+]/CD11c[+]), but not in the CD11b[-]/CD11c[+] population, which comprises cDC1 and plasmacytoid DC (*Figure 6A*). Interestingly, the exposure of PS residues was abrogated in Traf6[Δ/Δ] LN cDC (*Figure 6A*). Given that Traf6 deficiency does not modify the survival of hematopoietic cells or progenitors (*Fang et al., 2018*; *Figure 6—figure supplement 1A*), our observations on annexin-V binding indicate that Traf6 mediates the process of vesicle exocytosis. Having demonstrated that LPS/*Traf6* signaling is required for chemokine traffic/secretion in LN mature myeloid lineage cells, and NF-κB transcriptional activation is dispensable for LPS-dependent myeloid progenitor migration, we hypothesized that Traf6 acts through non-NF-κB-dependent IKK activity. To determine whether canonical LPS/TLR downstream effectors were involved in the process of myeloid progenitor migration, we analyzed the chemotaxis of BM myeloid progenitors toward a gradient generated by LN cells in the presence of LPS and specific inhibitors for interleukin receptor-associated kinase 1/4 (Irak1/4), ubiquitin-conjugating enzyme 13 (Ubc13), and IKKβ (*Figure 6B*). Increased myeloid progenitor migration was reversed by all three specific inhibitors (*Figure 6B*), indicating that the integrity of canonical signaling pathway upstream of NF-κB might be required to attract myeloid progenitors from the BM to the LN. The Traf6/IKK-dependent rapid response to LPS strongly suggests that LPS induces secretion of Ccl19 through a mechanism of rapid release from pre-stored pools. The release of pre-formed cytokines in pre-pooled, stored late endosomes depends on IKK activity through the phosphorylation of mediators of cell membrane fusion. SNAP23 is an essential component of the high-affinity receptor that is part of the general membrane fusion machinery and an important regulator of transport vesicle docking and fusion (*Karim et al., 2013*; *Suzuki and Verma, 2008*). Phospho-SNAP23(Ser95) is significantly upregulated by LPS in LN cDC (*Figure 6C*). Each of the inhibitors for Irak1/4, Ubc13, and IKK abrogated the activation of SNAP23 (*Figure 6C*, *Figure 6—figure supplement 1B*). Altogether, this set of data indicates that the activation through Traf6/Irak1/4/Ubc13 induced by LPS activates vesicular fusion and vesicular cargo release of pre-formed Ccl19 accumulated in late endosomes of LN myeloid cells. Analysis of steady-state LN myeloid cell populations identified a subpopulation of cDC but not pDC or macrophages, containing most of the cytoplasmic expression of Ccl19. Further analysis of the subpopulations of LN cDC2 demonstrated that B220[-]/CD8[-] cDC that expressed the endocytic receptor DEC-205 (CD205[+]) and the mannose receptor signal regulatory protein α (SIRPα, CD172a[+]) distinctly stored high levels of cytosolic Ccl19 (*Figure 6D*, *Figure 6—figure supplement 1C*) unlike other DC populations, which expressed low levels of intracellular Ccl19 (*Figure 6—figure supplement 1D*). These data indicate that a subpopulation of cDC2 stores intracellular Ccl19 and is potentially able to self-regulate the migration of its own progenitors in inflammation.

## Discussion

This study describes a previously unrecognized, rapid, emergent traffic of myeloid progenitor cells from the BM via lymphatic vessels directly to lymphatic tissues that bypass the peripheral blood stream. Careful analysis of serial femoral sections has not unveiled the existence of a communication

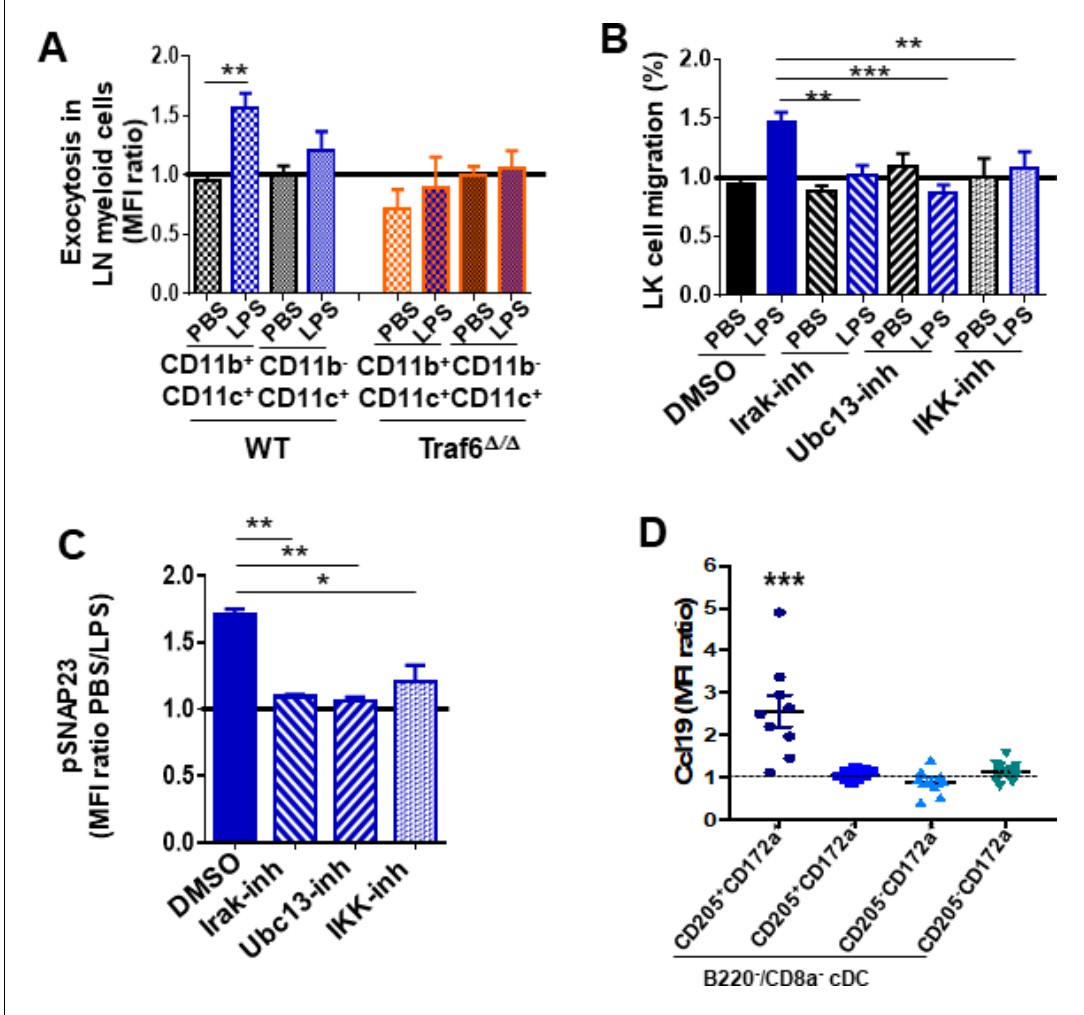

**Figure 6.** In vivo analysis of Ccl19/Ccr7 axis during inflammation. Pharmacological regulation of lipopolysaccharide (LPS)/Toll-like receptor (TLR) signaling pathway. (**A**) Annexin-V binding to membrane phosphatidylserine (PS) on lymph node (LN) myeloid populations from wild-type (WT) (left mosaic bars) and *Traf6*$^{\Delta/\Delta}$ (right mosaic bars) (n = 4 mice per group) after PBS or LPS administration. LN suspension cells were stained for myeloid surface markers including annexin-V and analyzed by flow cytometry. (**B**) Transwell migration of low-density bone marrow (LDBM)-derived LK cells toward gradient generated by pre-treated LN cells with dimethylsulfoxide (DMSO) (solid bars) as vehicle control and inhibitors (mosaic bars) against Irak1/4 (right lined), Ubc13 (left lined), and IKK (white squares), and following TLR signaling pathway activation by PBS (black) or LPS (blue). (**C**) Analysis of SNAP23 phosphorylation (Ser95) in LN myeloid cells previously treated with DMSO (solid bar) as vehicle control, Irak1/4 (right lined mosaic bar), Ubc13 (left lined mosaic bar), or IKK (white squares mosaic bar) inhibitors. Values represent two independent experiments as mean ± SD of two independent experiments performed in triplicate. (**D**) Mean fluorescence intensity (MFI) quantification of pre-stored Ccl19 into LN-residing CD11b$^{low}$/CD11c$^+$ cDC from non-manipulated mice by flow cytometry. Values represent mean ± SD of two or three independent experiments. p<0.05, **p<0.01, ***p<0.001.

The online version of this article includes the following figure supplement(s) for figure 6:

**Figure supplement 1.** Traf6 does not associate with changes in survival of hematopoietic cells, and progenitors and Ccl19 can be detected in a specific subpopulation of lymph node (LN) conventional dendritic cells (cDCs).

between lymphatic and blood vessels in BM further, suggesting the lack of communication between both circuits within the BM cavity and thus likely functional regional independence of each circuit. Our data thus also supports the recently described existence of functional lymphatic vessels in the bone. High-resolution confocal and multiphoton microscopy demonstrated the existence of Lyve1 + cells in which their transgenic reporter illuminated upon exposure to high-dose LPS in vivo along with tiny projections of lymphatics penetrating into the bone. Probably, bone processing and cleaning before fixation and decalcification may have deprived us (and other investigators) from a

better identification of notable, anatomically identifiable lymphatic vessels within the network of transcortical capillaries (*Grüneboom et al., 2019*). Although the exact anatomical and functional nature of this lymphatic network in the bone remains poorly defined, our data demonstrate that the lymphatic-mediated traffic from BM is highly activated in endotoxic inflammation and drains into regional LN chains.

Bone is a dynamic organ in constant remodeling. Upon inflammation, for example, cytokines and microbial LPS are capable of initiating bone absorption by activating osteoclasts (*Hardy and Cooper, 2009*; *Nason et al., 2009*). Systemic inflammation has been associated with osteoclast activation and osteoblast thinning (*Hardy and Cooper, 2009*; *Nason et al., 2009*), and bone lymphatic endothelial cells have been shown to arise rapidly from pre-existing regional lymphatics upon osteoclast activation (*Hominick et al., 2018*; *Monroy et al., 2020*). Osteoclast activation and osteoblast thinning are likely to facilitate transcortical migration of cells and fluid through existing transcortical vessels.

Our data showed that as early as 90 min after LPS administration myeloid progenitors to or are in closer proximity to the lymphatic endothelium in BM while 1.5 hr later there is an ~70% reduction of myeloid progenitors within the BM and a marked increase of myeloid progenitors in the LN tissue. This observation, along with the need of a longer period of time to detect an increase in the frequency of myeloid progenitors in peripheral blood, suggests that two temporally distinct waves of progenitors take place, a fast one to the lymphatic circulation followed by a slower one into the blood stream.

In our study, by using transgenic animals, we demonstrated that the administration of a single dose of LPS suffices to induce migration of GMP/MDP while no other types of progenitors or stem cells migrate to lymphatics in this first wave of egression before any significant contribution from or to PB and differentiate into short-lived LN cDCs in a murine acute model of inflammatory signaling by LPS, and that therefore they may modulate the course of infectious diseases and other inflammatory conditions. This traffic is also likely to happen in homeostatic conditions as previously shown (*Waskow et al., 2008*) while our analysis provides compelling evidence on its striking activation upon inflammation/LPS administration.

Our data support the migration of a distinctly immature progenitor population composed of GMP/MDP with ability to generate cDC in LN upon traffic from BM to LN. This traffic of myeloid progenitors from BM to LN is rapid, before LPS induces proliferation and/or apoptosis (*Nagai et al., 2006*; *Zhao et al., 2014*), and can be recapitulated in mice receiving intrafemoral adoptive transfer of GMPs. These GMP/MDP tend to localize in the T-cell areas of LN. Cheong et al. reported that migratory monocyte-derived cDC2 can also localize in T-cell areas of the LN and acquire an inflammatory phenotype DC-SIGN/CD209a[+] (*Cheong et al., 2010*). No significant mobilization of M-CSF-responding monocyte progenitors (CFU-M) can be found as early as 3 hr after LPS administration before any significant effect of LPS on proliferation may take place. Interestingly, the interference of this traffic by blocking Traf6-dependent signaling in Lyz2-expressing myeloid cells results in increased animal death. Ccr7[+] GMP/MDP, but not other myeloid or lymphoid progenitors, egress BM. Such egress follows differential tissue levels of Ccl19 resulting from activation of the secretion of the pre-formed chemokine by LN Lyz2-expressing mature myeloid cells, specifically a subpopulation of cDC expressing CD205 and CD172a. This process seems to be independent of Cxcl12 levels since no changes in Cxcl12 levels in LN, PB, or BM were observed, and this effect seems to be exclusively dependent on activation of non-canonical Traft6/IKK activity without need for transcriptional activation.

*Schmid et al., 2011* demonstrated that a population of common dendritic progenitors, a non-GMP-derived population of progenitors, can also migrate from the BM to lymphoid and non-lymphoid tissues in response to TLR agonists and generate both cDCs and pDCs (pDCs). The type of migration though depended on combined downregulation of Cxcr4 and upregulation of Ccr7, which seem to imitate the mechanism of GMP migration. Interestingly, Ccl21, which is expressed by lymphatic endothelial and stromal cells but not by myeloid cells (*Eberlein et al., 2010*), does not induce any differential gradient of secretion between LN and BM or blood, suggesting that Ccl21 may not be a primary mediator of the myeloid progenitor migration from BM to LN upon LPS challenge, while the hematopoietic deficiency of Ccl19 suffices to completely abrogate the mobilization of myeloid progenitors to LN induced by LPS.

Our data support the existence of a steady-state LN population of cDC that co-expresses the maturation antigens CD205 and CD172a and stores high levels of Ccl19 in their cytoplasm. An interesting possibility is, as our data indicate, that upon bacterial antigen challenge, differentiated myeloid cells of LN like cDCs, which respond to LPS by secreting chemokine-containing pre-formed exosomes, accelerate a positive feedback activation loop to recruit cDC progenitors to the lymphatic tissue. cDC in LNs might thus act as sensors for the presence of bacterial products and release Ccl19 within minutes. Individual DCs have a short half-life (1.5–2.9 days) (Kamath et al., 2002), and DC precursors have a short half-life in blood circulation (Breton et al., 2015). We posit that the migration of DC progenitors through the lymph tissues provides a direct afferent communication between the LN mature cDC population responsible for the secretion of the chemokine Ccl19 and at the same time allows the emergent migration of functional cDC progenitors from the BM to differentiate into lymphatic APCs.

Finally, our data also support the key role of an alternative inflammatory signaling pathway elicited by coordination of Traf6/IKK responsible for SNAP23 phosphorylation and Ccl19 secretion, before resulting in transcriptional regulation by their downstream effector NF-κB. Traf6 has been identified as a signaling molecule that can regulate splicing of downstream targets without affecting NF-κB in hematopoietic stem cells and progenitors (Fang et al., 2017). Our data further identifies non-canonical signaling pathways elicited by Traf6 in differentiated myeloid cells to modulate the migration of hematopoietic progenitors. Traf6-dependent, cytosolic-mediated IKK signaling allows a fast Ccl19 secretory response before inflammatory transcriptional and post-transcriptional signatures are expressed.

In summary, we describe, upon inflammation, a rapid trafficking of cDC-biased myeloid progenitors from the BM, via lymphatic vessels, directly to lymphatic tissues that bypasses the blood stream. This GMP/MDP migration represents a mechanism for fast replenishment of cDCs in lymphatic tissues. Rapid replenishment of a reserve of cDC-biased progenitors in LN may represent a major homeostatic function of this novel lymphatic circuit and may explain why the circulation of myeloid progenitors is conserved during the postnatal life.

## Materials and methods

### Mice

CD57Bl/6 (CD45.2$^+$) mice were used between 8–10 weeks of age and were purchased from Jackson Laboratory, Bar Harbor, ME; Harlan Laboratories, Frederick, MD. Mx1Cre;*Traf6*-floxed mice were generated by breeding Mx1-Cre transgenic mice (Mikkola et al., 2003) with biallelic *TRAF6* floxed mice (kindly provided by Dr. Yongwon Choi, University of Pennsylvania) (Kobayashi et al., 2003). Full chimeric mice were generated by non-competitive transplantation of Mx1Cre;WT or Mx1Cre;*Traf6*$^{flox/flox}$ whole BM cells into lethally irradiated B6.SJL$^{Ptprca\ Pepcb/BoyJ}$ (CD45.1$^+$) mice obtained from the CCHMC Animal Core. *Traf6* was deleted upon induced expression of Cre recombinase after 3–6 intraperitoneal injections (10 mg/kg/b.w. Poly(I:C); Amersham Pharmacia Biotech, Piscataway, NJ) every other day at 6 weeks after transplantation generating WT and Traf6$^{Δ/Δ}$ mice. *Lyz2-*$^{Cre}$;*Traf6*$^{flox/flox}$ mice were generated by non-competitive transplantation of *Lyz2*$^{Cre}$;WT or *Lyz2*$^{Cre}$;*Traf6*$^{flox/flox}$ whole BM cells into lethally irradiated B6.SJL$^{Ptprca\ Pepcb/BoyJ}$ (CD45.1$^+$) mice obtained from the Division of Experimental Hematology/Cancer Biology of Cincinnati Children's Hospital Research Foundation (CCHRF). Vav-Cre;*Traf6*$^{flox/flox}$ mice and their control counterparts were generated as previously described (Fang et al., 2018).

*Lyve1*$^{eGFP}$ (Pham et al., 2010) and β-actin-eGFP (Okabe et al., 1997) and *CX$_3$CR-1*$^{GFP}$ (Jung et al., 2000) transgenic mice were purchased from Jackson Laboratories. C57BL/6 mice for circadian cycle analysis of CFU-C were maintained on a 14 hr light/10 hr darkness lighting schedule. This study was performed in strict accordance with the recommendations in the Guide for the Care and Use of Laboratory Animals of the National Institutes of Health. All of the animals were handled according to approved Institutional Animal Care and Use Committee (IACUC) protocol #2019-0041 of Cincinnati Children's Hospital.

## Human specimens

Lymphadenopathies from patients were obtained through Institutional Review Board-approved protocols of the Hospital Reina Sofia (Cordoba, Spain), donor-informed consent, and legal tutor approval in the case of patients younger than 18 years old. Diagnostic lymphadenopathy biopsies from 64 consecutive patients from 2009 until 2013 were analyzed in this study. The median age of patients was 34 years old (range: 3–89). Diagnosis and histological classification of the type of lymphadenopathy and tumors were based on previously published criteria (*Campo et al., 2011*; *Weiss and O'Malley, 2013*). Anatomical location of lymphadenopathies is described in *Supplementary file 1*. Specimens were blindly analyzed through adjudication of unique identifiers.

## LPS injection and samples collection

Mice received a single intraperitoneal injection of 30 mg/kg of *E. coli* LPS (Sigma-Aldrich, St Louis, MO) or PBS as vehicle control and were executed always at ZT4 or 4 hr after the initiation of light into the animal room. At different time points after PBS or LPS administration, BM cells from femurs, tibias, and pelvis were harvested by crushing in PBS containing 2% of FBS and erythrocytes were lysed using a hypotonic buffer from BD Biosciences. Blood was collected by retro-orbital bleeding or cardiac injection. Liver, kidney, and thoracic duct cells were harvested by enzymatic digestion solution with collagenase II (1 mg/mL, Thermo Fisher–Gibco, Waltham, MA) and dispase (5 mg/mL, Gibco, Life Technologies) in a shaking water bath at 37°C for 1 hr. Spleen cells were isolated by scraping with slides in sterile PBS following red blood cell (RBC) lysis (Pharm Lyse; BD Biosciences, San Jose, CA). Extracted LN were derived from the cervical and axillary chains exclusively.

## Myeloid progenitor counts assay in vitro

Cells from BM, LNs, thoracic duct, blood, spleen, liver, and kidneys were depleted of RBC by 2 min incubation in Pharm Lyse (BD Biosciences), washed, counted, and plated in SDisolid methylcellulose media (Methocult 3434; StemCell Technology, Vancouver, Canada) and cultured in an incubator (37°C, 5% $CO_2$/>95% humidity), and the number of CFU-C was scored on day 7 or 8 of culture using an inverted microscope. To examine the type of myeloid progenitors migrating into LN, we used base methylcellulose medium (Methocult 3134; StemCell Technology) supplemented with 30% FCS, 1% protease-free, deionized BSA (Roche), 100 mM b-mercaptoethanol, 100 IU/mL penicillin, 0.1 mg/mL streptomycin, and any of the following: for CFU-GM, rm-GM-CSF (100 ng/mL, PeproTech, Rocky Hill, NJ) for specific analysis of CFU-GM content, rm-M-CSF (100 ng/mL, PeproTech ) for specific analysis of CFU-M content, or rh-G-CSF (100 ng/mL, Neupogen) for specific analysis of CFU-G content.

## Long-term competitive repopulation assay

To analyze the long-term reconstitution capacity of HSC/Ps mobilized into LNs after LPS administration at different time of periods, $4-5 \times 10^6$ of erythrocyte-depleted CD45.2$^+$ LN suspension cells were prepared in sterile conditions and transplanted together with $2.5 \times 10^5$ CD45.1$^+$ BM competitor cells into lethally irradiated CD45.1$^+$ recipient mice. In some experiments, $10^4$ LN SP cells or $10^3$ BM SP cells from CD45.2$^+$ mice were competitively transplanted into CD45.1$^+$ recipient mice. Competitive repopulating units (CRU) analysis was performed by flow cytometry analysis (BD Biosciences) at different time points post-transplantation (*Harrison, 1980*).

## Flow cytometry analysis and cell sorting

For immunophenotype analysis of HSC/P populations by fluorescence-activated cell sorter (FACS), erythrocyte-depleted BM cells were stained first for lineage markers with biotin-labeled mouse lineage panel (BD Biosciences, Pharmingen) containing anti-CD3e (CD3ε chain), anti-TER-119/erythroid cells (Ly-76), anti-Gr1 (Ly6G and Ly-6C), anti-CD45R (B220), anti-CD11b (integrin α chain, Mac1α), followed by allophycocyanin and cyanine dye Cy7-(APC-Cy7)-conjugated streptavidin, allophycocyanin (APC)-conjugated anti-c-Kit (clone 2B8), R-phycoerythrin and cyanine dye Cy7 (PECy7)-conjugated anti-Sca1 (clone D7), eFluor 450-conjugated anti-CD34 (clone RAM34) (Affymetrix eBioscience, San Diego, CA), and PerCP and cyanine dye Cy5.5 (PerCP Cy5.5)-conjugated anti-Fcγ-RII/III (clone 2.4G2) (BD Biosciences). FACS sequential discrimination on a lineage negative gated population was used to identified LK myeloid progenitor subpopulations: Lin$^-$,c-Kit$^+$Sca1$^-$CD34$^+$

FcγRII/III$^+$ (GMP); Lin$^-$,c-Kit$^+$Sca1$^-$CD34$^+$FcγRII/III$^{lo}$ (CMP); Lin$^-$ cKit$^+$Sca1$^-$ CD34$^+$ FcγRII/III$^-$. LSK (Lin-Sca1 +c Kit-) subpopulations were distinguished as Lin$^-$,c-Kit$^-$Sca1$^+$ CD34$^-$Flt3$^-$ for LT-HSC Lin$^-$,c-Kit$^-$Sca1$^+$CD34$^+$Flt3$^-$ for ST-HSC, and Lin$^-$,c-Kit$^-$Sca1$^+$CD34$^+$Flt3$^+$ for MPPs. For chimera analysis in repopulated animals, 20 μL of red cell-depleted blood was stained with fluorescein isothiocyanate (FITC)-conjugated anti-CD45.1 (clone A20), R-phycoerythrin and cyanine dye Cy7 (PECy7)-conjugated anti-CD45.2 (clone 104), allophycocyanin (APC)-conjugated anti-CD11b (clone M1/70), allophycocyanin and cyanine dye Cy7-(APC-Cy7)-conjugated anti-B220 (clone RA3-6B2), R-phycoerythrin (PE)-conjugated anti-CD3e (clone 145-2 C11) and BD Horizon V450-conjugated anti-Gr1 (clone RB6-8C5), PerCPefluor710 anti-CD115 (clone AFS98), and R-phycoerythrin-conjugated anti-CD135 or anti-Flt3 (clone A2F0.1). All monoclonal antibodies were purchased from BD Biosciences, Pharmingen. Cell acquisition was performed by flow cytometry (LSRFortessa I, BD Biosciences) equipped with FACSDIVA software (BD Biosciences) for multiparameter analysis of the data. FACS strategies were CD45.1$^-$CD45.2$^+$CD3ε$^+$B220$^-$CD11b$^-$ for LN-T cells, CD45.1$^-$ CD45.2$^+$CD3ε$^-$B220$^+$/CD11b$^-$ for LN-B cells, and CD45.1$^-$ CD45.2$^+$CD3ε$^-$/B220$^-$/CD11b$^+$ for LN myeloid cells in a FACSAria II cell sorter (BD Biosciences). For BM and LN -SP cells analysis and sorting, 2 × 10$^6$ cells/mL were stained with Hoescht 3342 (5 μg/mL) as described previously (*Cheong et al., 2010*). For intracellular analysis of the phosphorylated state of SNAP23 protein, surface antigen-labeled cells were fixed with Cytofix buffer (BD Biosciences) for 20 min and then permeabilized using Cytofix/Cytoperm buffer (BD Biosciences) for 20 min. After washing, cells were stained intracellularly using a rabbit non-conjugated monoclonal anti-phospho-SNAP23(Ser$^{95}$) (*Karim et al., 2013*) for 40 min in Perm/Wash Buffer 1x (BD Biosciences) with 0.5% of rabbit serum. Cells were then incubated with a secondary Alexa Fluor 488-conjugated (Thermo Fisher–Invitrogen) goat anti-rabbit antibody for 40 min in Perm/Wash Buffer 1x with 0.5% of goat serum. All incubations after cell stimulation were done on ice and in darkness. Single-cell analysis was performed by flow cytometry and the histogram-overlay graphed (LSRFortessa I; FlowJo xV0.7 software; BD Biosciences). The mean fluorescence intensity (MFI) ratio was calculated as the ratio of the fluorescence intensities of LPS-stimulated to PBS-stimulated (control).

## Vesicle exocytosis analysis

LN suspension cells from Mx1Cre;WT and Mx1Cre;Traf6$^{Δ/Δ}$ mice were obtained to performed LPS stimulation. 10$^6$ cells were plated into 24-well plates and treated with PBS or LPS for 1 hr. After 15 min labeling with surface antibodies against CD45.2 (clone 104), CD11c (clone HL3), CD11b (clone M1/70), and B220 (clone RA3-6B2), the samples were washed twice and then stained for annexin-V for 15 min and in darkness. All antibodies were purchased from BD Biosciences, Pharmingen. Single-cell analysis was performed using flow cytometry and the histogram-overlay graphed (LSRFortessa I; FlowJo xV0.7 software; BD Biosciences). The MFI ratio between LPS MFI and PBS MFI was calculated.

## Apoptosis analysis

To determine HSPC cell death, BM cells were isolated and incubated with biotin-conjugated lineage markers as described above, followed by staining with streptavidin eFluor450, Sca-1-PE-Cy7, c-Kit-APC-eFluor780. The cells were then fixed, permeabilized, and stained with TUNEL TMR red (12156792910, Roche). Analysis was performed using FACSCanto and/or LSRII flow cytometers and with either Diva or FlowJo software.

## Homing and seeding assays

For homing assays, 2 × 10$^6$ of Lin$^-$ cells, previously depleted by immunomagnetic selection (Lineage Cell Depletion kit, Miltenyi Biotec, Auburn CA), were stained by 1,1′-dioctadecyl-3,3,3′,3′,tetramethylindocarbocyanine perchlorate;CILC18(3) (5 μM/mL DiI, Thermo Fisher–Invitrogen) and adoptively transferred intravenously into non-myeloablated *Lyve1$^{eGFP}$* mice. 17 hr later, one single LPS dose (3 mg/mL) or vehicle control (PBS) was administered intraperitoneally. 3 and 6 hr later (ZT7 or ZT10), mice were euthanized with pentobarbital (60–80 mg/kg) and the whole body was fixed using a freshly made solution of PBS plus 2% of paraformaldehyde and 0.05% of glutaraldehyde infused by perfusion pump through left ventricle of the animal. 15–20 min later, the BM cells and LN organs

were harvested and the percentage of labeled Lin⁻ cells that had homed into BM was determined by FACS analysis. The homing calculation was done as previously reported (*Boggs, 1984*).

## Microscopy

Fixed LN organs were permeabilized for 15 min with PBS containing 0.2% of Triton X-100. To detect GFP on the lymphatic endothelium, LN were incubated overnight with a primary antibody anti-GFP+ (Thermo Fisher–Invitrogen). LNs were scanned by confocal microscopy (Nikon A1R GaAsP) through multidimensional acquisition to construct 3D representations of the whole organ at ×10 magnification. The merged images of GFP/DiI or DAPI/DiI are presented, and the total cell number of labeled Lin⁻ cells was counted manually. Finally, harvested femurs were decalcified for 14 days with 10% of EDTA (Sigma-Aldrich) in PBS and embedded in paraffin. Longitudinal sections of bone were cut to 4 µm thickness and were then de-paraffinized and broke the protein crosslink before stain by antigen retrieval treatment with citrate buffer pH 6 (*Cancelas et al., 2005*). Then bone sections were permeabilized with 0.2% of Triton X-100 for 15 min and blocked with 5% of BSA for 1 hr. Slides were stained with primary antibodies anti-GFP (chicken polyclonal, Abcam Inc, Cambridge, MA) and rat anti-mouse panendothelial cell antigen (clone, MECA-32, BD Biosciences, Pharmingen) at 4°C overnight. Then we stained with secondary antibodies, goat anti-rat Alexa Fluor-488 and goat anti-chicken Alexa Fluor-568, all from Invitrogen at 1:1000 v/v concentration for 1 hr at room temperature. Blood and lymphatic vessels were scanned by confocal microscopy (Nikon A1R GaAsP) through multidimensional acquisition to construct 3D representation.

To further characterize lymphatic system in bone tissue and BM cavity and to image the close proximity of homed Lin⁻/DiI⁺ to lymphatic vessels into *Lyve1^eGFP* mice, we utilized multiphoton IVM as previously described (*Gonzalez-Nieto et al., 2012*; *Köhler et al., 2009*). After LPS/PBS injections, long bones were harvested and muscle were carefully cleaned. Further, bone tissues were cautiously trimmed with an electric drill (Dremel) to get better access of the BM cavity for imaging by leaving a very thin (~30–40 µm) layer of bone tissue. Bones were mounted in 2% low- melting agarose to minimize movements during imaging and covered with PBS. Multiphoton microscopy on the long bones (femur and tibia) was subsequently performed using a Nikon A1R Multiphoton Upright Confocal Microscope equipped with Coherent Chameleon II TiSapphire IR laser, tunable from 700 to 1000 nm, and signal was detected by low-noise Hamamatsu photomultiplier (PMT) tubes. Bone tissue was identified as second-harmonic (SHG) signal (PMT). Bones were images in PBS using a 25× Apo 1.1 NA LWD water Immersion objective and NIS image software. For initial standardization, bones were scanned at wavelength of 800, 850, and 900 nm detecting GFP (530 nm) and DiI red (580 nm). For imaging, a 500 × 500 µm area was scanned in ~35 steps of 4 µm down to 120–150 µm depth using an illumination wavelength of 800 nm detecting SHG signal (480 nm), green (530 nm), and red (580 nm) fluorescence. Control C57BL/6 mice were used as a negative control for *Lyve1^eGFP* mice to detect specific signal for GFP-lymphatic system in bone tissue and BM cavity. Lymphatic vessels were well detected in the bone tissue using *Lyve1^eGFP* mice with DiI-labeled Lin⁻ cells in the BM cavity. For quantification of proximity of Lin⁻/DiI⁺ with lymphatic system, Imaris software was used to measure distance between DiI-labeled cells and GFP-positive lymphatic vessels using 3D images.

## Analysis of L-selectin dependence of femoral GMP/MDP migration to regional LN

C57Bl6 mice received single intraperitoneal injections of MEL14 (CD62L) antibody (BioXcell) 200 µg. Control mice received the same amount of Rat IgG2a. Post 3 days of MEL14 antibody treatment, LDBM cells from ß-actin eGFP mice were injected intrafemorally. LPS (30 mg/kg, BW) was injected post 1 hr of interfemoral injections. Mice were sacrificed at 3 hr post-administration, and ipsilateral and contralateral regional and distant LNs (inguinal, popliteal, axillary, and cervical) were isolated for analysis. LN cells were stained for granulocyte-macrophage progenitor (GMP) markers and anti-CD19-PECy7 (Cat# 552854, BD Biosciences) and analyzed by flow cytometry and quantified the GFP⁺ GMP and B lymphocyte populations migrating to LN.

## Femoral GFP⁺ progenitor migration in WT and Ccl19⁻/⁻ hematopoietic chimeras

Hematopoietic chimeras of WT and Ccl19⁻/⁻ (Link et al., 2007) BM cells were generated by transplantation into CD45.1 + mice, similarly to Mx1Cre;d hematopoietic chimeras. Mice were followed for 8 weeks and found to have >95% chimera of CD45.2+ cells in peripheral blood. After 8 weeks, femoral LDBM cells from donor congenic-actin transgenic, CD45.2+ mice were injected ($5 \times 10^5$ per mouse) intrafemorally to both WT and Ccl19 hematopoietic chimeric mice. PBS or LPS (30 mg/kg, b. w.) were injected at 1 hr post-intrafemoral injections and sacrificed at 3 hr post-administration of LPS. At that time, ipsilateral LN from inguinal and popliteal regions was isolated. Suspension of LN cells was counted and stained with specific antibodies for GMP and MDP characterization, and the frequency of different GFP⁺ GMP and MDP populations was analyzed by flow cytometry as mentioned above.

## Chemotaxis/migration assays

For non-cell-autonomous effect analysis, $5 \times 10^5$ of BM or LN nucleated cells from Mx1Cre;WT *or* Mx1Cre;Traf6$^{\Delta/\Delta}$ CD45.2⁺ were layered on bottom wells of 24-well transwell plate (Corning Inc, Lowell, MA) together with 100 ng/mL of LPS, and $1 \times 10^5$ WT CD45.1⁺ LDBM cells were layered on upper chamber at 37°C, 5% $CO_2$. For cell-autonomous effect analysis, $5 \times 10^5$ of BM or LN nucleated cells from WT CD45.1⁺ mice were layered in the lower chamber with 100 ng/mL of LPS, and $1 \times 10^5$ Mx1Cre;WT or Mx1Cre;Traf6$^{\Delta/\Delta}$ CD45.2⁺ LDBM cells were layered in the upper chamber at 37°C, 5% $CO_2$. After 4 hr, cells were resuspended and those adhered to the bottom layer were collected using an enzyme-free cell dissociation buffer (Cell Dissociation Buffer, enzyme free, PBS, Thermo Fisher–Gibco). Progenitor responses toward migratory gradient were analyzed by flow cytometry analysis of LK cells. The percentage of migration was calculated by dividing the number of LK in the outputs by the number of LK in the inputs and multiplied by 100. PBS was included as negative control. All assays were performed in triplicate.

## NF-κB activity repression in myeloid cells

To analyze the NF-κB-dependent or -independent mechanism of myeloid progenitor migration, BM Lin⁻ were transduced with pMSCVpuro-eGFP bicistronic retroviral vector encoding the full length of IκBα mutant (super-repressor) in the presence of the recombinant fragment of fibronectin, CH296 (Takara Bio Inc, Madison, WI) for 16 hr at 37°C. 24 hr later, GFP⁺ cells were sorted and macrophages were generated (Chang et al., 2014). To characterize the expanded population, R-phycoerythrin (PE)-conjugated anti-CD169 (clone 3D6.112), PerCP-efluor 710-conjugated anti CD115 (clone AF598) (Affymetrix eBioscience), allophycocyanin and cyanine dye Cy7-(APC-Cy7)-conjugated anti-CD11b, efluor 450-conjugated anti-F4/80 (clone BM8) (Affymetrix eBioscience), and Alexa Fluor 647-conjugated anti-CD68 (clone FA-11) (BD Biosciences) were used for FACS analysis. $50 \times 10^3$ differentiated and transduced macrophages with empty or IκBα super-repressor were layered on bottom wells of 24-well transwell plate in the presence of 100 ng/mL of LPS, and $1 \times 10^5$ WT LDBM cells were layered on upper chamber at 37°C, 5% $CO_2$. 4 hr later, migrated LK cells were determined by flow cytometry as described above. All assays were performed in triplicate.

## Secretome and individual cytokine/chemokine analyses

BM, plasma, and LN cells were isolated in PBS containing a protease inhibitor cocktail (Roche Diagnostics, Chicago, IL), and Ccl19/Ccl21 levels were determined by indirect sandwich of enzyme-linked immunosorbent assay (ELISA) following manufacturer's instructions (R&D Systems, Minneapolis, MN). Multi-analytic profiling beads using Milliplex Multiplex mouse cytokine/chemokine panels (EMD Millipore, Billerica, MA) according to manufacturer's instructions were used to analyze chemokines and cytokines profile into BM and LN tissues at different time periods after LPS or PBS administration into WT mice.

## In vivo administration of anti-Ccr7 and anti-Ccl19

Monoclonal rat IgG2a antibody specific for Ccr7 (clone 4B12) or polyclonal goat IgG antibody for Ccl19 (AF880) and control rat IgG2a or control goat purified IgG were obtained from R&D Systems.

50 µg of antibodies were injected twice into C56BL/6 mice within 15 hr (first dose i.v. and the second dose i.p.).

## Small-molecule inhibitors

The Irak1/4 inhibitor I, ubiquitin-conjugating enzyme E2 N (UBE2N) inhibitor, Ubc13 inhibitor (*Rhyasen et al., 2013*), and IκB kinase inhibitor (PS-1145 dihydrochloride) were purchased from Sigma-Aldrich. LN cells from C57BL6 mice were treated with 10 µM of IRAK-Inh, 0.2 µM of Ubc13-Inh, and 10 µM of IKK-Inh for 45 min and compared with the vehicle dimethylsulfoxide (DMSO) at 0.1% in PBS. Monensin (eBioscience) was used at 2 µM.

## Statistical analysis

Quantitative data is given as mean ± standard error of the mean (SEM) or standard deviation (SD). Statistical comparisons were determined using an unpaired Student's t-test, non-parametric Mann–Whitney test, and one-way or two-way ANOVA with Bonferroni corrections. A value of $p < 0.05$ was considered to be statistically significant.

## Acknowledgements

We thank Dr. Andre Olsson for flow cytometry analysis advice and Ms. Margaret O'Leary for editing manuscript. We also want to thank Dr. Andres Hidalgo (CNIC, Madrid, Spain, and Yale University, New Haven, CT), and Drs. Daniel Lucas, Leighton Grimes, Michael Jordan, Edith Jansen, and Jizhou Zhang (Cincinnati Children's Hospital Medical Center) for helpful discussions on data interpretation and experimental approaches. This project has been funded by the Junta de Andalucía of Spain (JS-L) and NIH R01 GM110628 and DK124115 grants (JAC). The authors declare they have no relevant conflicts of interest. We want to thank Francisco M Gutierrez for human CFU-C pictures. We also want to thank Jeff Bailey and Victoria Summey for technical assistance and the Mouse and Research Flow Cytometry Core Facilities, both supported by the NIH/CEMH grant P30DK090971-01.

## Additional information

### Funding

| Funder | Grant reference number | Author |
| --- | --- | --- |
| National Institutes of Health | GM110628 | Jose A Cancelas |
| National Institutes of Health | DK124115 | Jose A Cancelas |
| Junta de Andalucía | | Juana Serrano-Lopez |

The funders had no role in study design, data collection and interpretation, or the decision to submit the work for publication.

### Author contributions

Juana Serrano-Lopez, Conceptualization, Data curation, Formal analysis, Funding acquisition, Validation, Investigation, Visualization, Methodology, Writing - original draft, Writing - review and editing; Shailaja Hegde, Data curation, Formal analysis, Validation, Investigation, Visualization, Methodology, Writing - original draft, Writing - review and editing; Sachin Kumar, Data curation, Formal analysis, Validation, Investigation, Visualization, Methodology, Writing - original draft; Josefina Serrano, Resources, Investigation; Jing Fang, Resources, Methodology; Ashley M Wellendorf, Investigation, Methodology; Paul A Roche, Resources; Yamileth Rangel, Data curation, Investigation, Methodology; Leolene J Carrington, Resources, Validation, Methodology; Hartmut Geiger, Resources, Supervision, Investigation, Methodology, Writing - review and editing; H Leighton Grimes, Sanjiv Luther, Resources, Writing - review and editing; Ivan Maillard, Resources, Supervision, Validation, Methodology, Writing - review and editing; Joaquin Sanchez-Garcia, Conceptualization, Resources, Supervision, Investigation, Project administration, Writing - review and editing; Daniel T Starczynowski, Conceptualization, Resources, Supervision, Project administration, Writing - review and editing; Jose A Cancelas, Conceptualization, Resources, Data curation, Formal analysis, Supervision, Funding

acquisition, Investigation, Methodology, Writing - original draft, Project administration, Writing - review and editing

## Author ORCIDs
Leolene J Carrington http://orcid.org/0000-0002-7352-3270
Hartmut Geiger http://orcid.org/0000-0002-5794-5430
Jose A Cancelas https://orcid.org/0000-0002-1291-7233

## Ethics

Human subjects: Lymphadenopathies from patients were obtained through Institutional Review Board-approved protocols of the Hospital Reina Sofia (Cordoba, Spain), donor informed consent and legal tutor approval in the case of patients younger than 18 years old. Specimens were blindly analyzed through adjudication of unique identifiers.

Animal experimentation: This study was performed in strict accordance with the recommendations in the Guide for the Care and Use of Laboratory Animals of the National Institutes of Health. All of the animals were handled according to approved institutional animal care and use committee (IACUC) protocol #2019-0041 of Cincinnati Children's Hospital.

## Decision letter and Author response

Decision letter https://doi.org/10.7554/eLife.66190.sa1
Author response https://doi.org/10.7554/eLife.66190.sa2

## Additional files

### Supplementary files

• Supplementary file 1. Histological categorization of human lymph node biopsies and their side population (%) frequency.

• Transparent reporting form

## Data availability

All data generated or analysed during this study are included in the manuscript and supporting files.

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

# Appendix 1

**Appendix 1—key resources table**

| Reagent type (species) or resource | Designation | Source or reference | Identifiers | Additional information |
|---|---|---|---|---|
| Genetic reagent (*Mus musculus*) | C57Bl/6 | Harlan Labs. 044 | RRID:MGI: 2161078 | C57BL/6NHsd; CD45.2+ |
| Genetic reagent (*M. musculus*) | B6.SJL$^{Ptprca\ Pepcb/BoyJ}$ | Jackson Labs 002014 | RRID:MGI:109863 | CD45.1+ |
| Genetic reagent (*M. musculus*) | Mx1-Cre | Jackson Labs 003556 | RRID:MGI: 4358794 | Tg(Mx1-cre)1Cgn |
| Genetic reagent (*M. musculus*) | Traf6$^{flox/flox}$ | PMID:29386112 | | Provided by Dr. Daniel Starczynowski, CCHMC, Cincinnati, OH |
| Genetic reagent (*M. musculus*) | *Lyz2$^{Cre}$* | PMID:10621974 | RRID:MGI: 1934631 | Mature myeloid cell promoter driving Cre recombinase expression |
| Genetic reagent (*M. musculus*) | Vav-Cre | PMID:14975238 | | Provided by Dr. Andrew Roberts, WEHI, Melbourne, Australia |
| Genetic reagent (*M. musculus*) | *Lyve1$^{eGFP}$* | Jackson Labs 012601 | RRID:MGI: 4421655 | Lyve1$^{tm1.1(EGFP/cre)Cys}$ Enhanced green fluorescent protein (EGFP) in these knock-in mice is driven by the *Lyve1* (lymphatic vessel endothelial hyaluronan receptor 1) promoter |
| Genetic reagent (*M. musculus*) | β-actin-eGFP | Jackson Labs, 003291 | RRID: MGI:2686773 | B6 ATCTb-EGFP Enhanced GFP (eGFP) cDNA under the control of a chicken β-actin promoter and cytomegalovirus enhancer, have widespread eGFP fluorescence, with the exception of erythrocytes and hair |
| Genetic reagent (*M. musculus*) | *CX$_3$CR-1$^{GFP}$* | Jackson Labs 005582 | RRID:MGI:J:84544 | Enhanced green fluorescent protein (*EGFP*) sequence replacing the first 390 bp of the coding exon (exon 2) of the chemokine (C-X3-C motif) receptor 1 (Cx3cr1) gene |
| Lipopolysaccharide | LPS | Sigma-Aldrich | Cat# L2630 | Biological source is *Escherichia coli* (O111:B4) Purified by phenol extraction Stimulation with this generates various pro inflammatory cytokines |
| Enzyme | Collagenase-II | Thermo Fisher Scientific, Gibco | Cat# 17101015 | Biological source is *Clostridium histolyticum*, digestion of bone, cartilage, tissues |
| Enzyme | Dispase | Thermo Fisher Scientific, Gibco. | Cat# 17105041 | Biological source is *Bacillus polymyxa* Helpful in dissociation of primary cells from the tissues |

*Continued on next page*

*Appendix 1—key resources table continued*

| Reagent type (species) or resource | Designation | Source or reference | Identifiers | Additional information |
|---|---|---|---|---|
| Lysing buffer | Pharm Lyse (red blood lysing buffer) | BD Biosciences | Cat# 555899 | Helpful in red blood cell lysis and results in good light scatter separation of lymphocytes and red blood cell debris when used in flow cytometry |
| Methylcellulose media | Methocult-GF M3434 | StemCell Technology | Cat# 03434 | Semisolid medium to growth murine HPCs |
| Base methylcellulose medium | Methocult-M3134 | StemCell Technology | Cat# 03134 | Incomplete medium that contains 2.6% methylcellulose in Iscove's MDM |
| Cytokine | Recombinant murine GM-CSF | PeproTech | Cat# 315-03 | Biological source is *E. coli* Hematopoietic growth factor stimulates macrophages and neutrophils Produced in endothelial cells, fibroblasts |
| Cytokine | Recombinant murine G-CSF | PeproTech | Cat# 250-05 | Biological source is *E. coli* Hematopoietic growth factor that stimulates the committed progenitor to neutrophils and improve the functional activities |
| Cytokine | Recombinant murine M-CSF | PeproTech | Cat# 315-02 | Biological source is *E. coli* It facilitates the hematopoietic recovery after the bone marrow transplantation |
| Cytokine | Human G-CSF | NEUPOGEN (filgrastim, Amgen) | | |
| Lineage cocktail antibody | Anti-CD3e (CD3ε chain), anti-TER-119/erythroid cells (Ly-76), anti-Gr1 (Ly6G and Ly-6C), anti-CD45R (B220), anti-CD11b (integrin α chain, Mac1α) | Pharmingen, BD Biosciences | 559971 RRID:AB_10053179 | Sorting out or depleting lineage expressing cells and enriching hematopoietic progenitors in bone marrow CD3e is biotin Hamster anti-mouse, dilution 1:200, CD45R is rat anti-mouse dilution 1:200, CD11b is rat anti-mouse, Ter119 is rat anti-mouse |
| Recombinant protein | Cy7-APC-Cy7-conjugated streptavidin | Pharmingen, BD Biosciences | 554063 RRID:AB_10054651 | |
| Antibody | APC-conjugated anti-c-Kit | Pharmingen, BD Biosciences | 553356 RRID:AB_398536 | Clone 2B8 Rat anti-mouse specifically binds to C-kit a transmembrane tyrosine kinase receptor Dilution 1:200 |
| Antibody | PECy7-conjugated anti-Sca1 | Pharmingen, BD Biosciences | 558162 RRID:AB_647253 | Clone D7 Rat anti-mouse Dilution 1:200 |

*Continued on next page*

*Appendix 1—key resources table continued*

| Reagent type (species) or resource | Designation | Source or reference | Identifiers | Additional information |
|---|---|---|---|---|
| Antibody | eFluor 450-conjugated anti-CD34 | Affymetrix, eBioscience | RRID:AB_2043838 | Clone RAM34 Monoclonal antibody reacts with mouse CD34 Dilution 1:200 |
| Antibody | PerCP Cy5.5-conjugated anti-Fcγ-RII/III | Pharmingen, BD Biosciences | 560540 RRID:AB_1645259 | Clone 2.4G2 It specifically recognizes a common non-polymorphic epitope on the extracellular domains of the mouse FcγIII Rat anti-mouse Dilution 1:200 |
| Antibody | FITC-conjugated anti-CD45.1 | Pharmingen, BD Biosciences | 553775 RRID:AB_395043 | Clone A20 Monoclonal antibody specifically binds to CD45.1 of all leukocytes Dilution 1:200 |
| Antibody | PECy7-conjugated anti-CD45.2 | Pharmingen, BD Biosciences | 560696 RRID:AB_1727494 | Clone 104 This recognizes CD45 on all leukocytes that of most mouse strains example: C57BL6, Balb/C, etc. Mouse monoclonal Dilution 1:200 |
| Antibody | APC-conjugated anti-CD11b | Pharmingen, BD Biosciences | Cat# 553312 RRID:AB_398535 | Clone M1/7 Specifically binds to CD11b or integrin alfa M Rat anti-mouse Dilution 1:200 |
| Antibody | -APC-Cy7-conjugated anti-B220 (clone RA3-6B2 | Pharmingen, BD Biosciences | 552094 RRID:AB_394335 | Clone RA3-6B2 Binds to an extracellular domain of the transmembrane CD45, which is expressed in all B lymphocytes Rat anti-mouse Dilution 1:200 |
| Antibody | PE-conjugated anti-CD3e | Pharmingen, BD Biosciences | 552774 RRID:AB_394460 | Clone 145-2 C11 Binds to T-cell receptor expressed CD3 complex that is expressed in thymocytes and mature T cells, etc. Hamster anti-mouse Dilution 1:200 |
| Antibody | BD Horizon V450-conjugated anti-Gr1 | Pharmingen, BD Biosciences | 560454 RRID:AB_1645285 | RB6-8C5 Rat anti-mouse, clonality unknown Dilution 1:200 |
| Antibody | PerCP efluor710 anti-CD115 | Thermo Fisher Scientific, eBioscience | 46-1152-82 RRID:AB10597740 | Clone AFS98 Monoclonal antibody against mouse Cd115, receptor of macrophage colony-stimulating factor Dilution 1:200 |
| Antibody | R-phycoerythrin conjugated anti-CD135 | BD Biosciences | Cat# 562537 RRID:AB_2737639 | A2F10.1 This is a monoclonal antibody specifically binds to Flk2-Flt3 Dilution 1:200 |

*Continued on next page*

*Appendix 1—key resources table continued*

| Reagent type (species) or resource | Designation | Source or reference | Identifiers | Additional information |
|---|---|---|---|---|
| DNA die | Hoechst 33342 | Fisher Scientific, Invitrogen | Cat# H3570 | This is a cell permeant nuclear counterstain that emits blue florescence when bound to dsDNA 10 mg/mL |
| Antibody | Rabbit non-conjugated monoclonal anti-phospho-SNAP23(Ser95) | Paul A. Roche | | Center for Cancer Research, National Cancer Institute, NIH |
| Fixation buffer | BD Cytofix | BD Biosciences | Cat# 554655 | For immunofluorescent staining of intracellular cytokines |
| Fixation/permeabilization Kit | BD Cytofix/Cytoperm | BD Biosciences | Cat# 554714 | This provides fixation and permeabilization of the cells |
| Perm/Wash buffer | BD Perm/Wash | BD Biosciences | Cat# 554723 | This is used in intracellular cytokine staining to permeabilize cells and can be used as antibody diluent |
| Antibody | Secondary Alexa Fluor 488-conjugated, goat anti-rabbit | Thermo Fisher Scientific, | RRID:AB_2633280 | Used as secondary antibody Goat polyclonal Dilution 1:500 |
| Software, algorithm | FlowJo xV0.7 | https://www.flowjo.com/solutions/flowjo | RRID:SCR_008520 | |
| In Situ Cell Death Detection Kit | TMR red | Sigma-Aldrich, Roche | Cat# 12156792910 | This is used to detect apoptosis on a single-cell level |
| Depletion kit | Lineage Cell Depletion | Miltenyi Biotech | Cat# 130-090-858 | Depletion of mature hematopoietic cells |
| Plasma membrane dye | Dil stain | Thermo Fisher Scientific, Invitrogen | Cat# D3911 | This is a lipophilic membrane stain that diffuses laterally to stain the entire cell |
| Detergent solution | Triton X-100 | Sigma-Aldrich | Cat# X100 | A mild detergent used to break the cell membrane |
| Antibody | Anti-GFP | Abcam | Cat# ab13970 RRID:AB_300798 | Chicken polyclonal antibody Dilution 1:500 |
| Antibody | Purified rat anti-mouse panendothelial cell antigen | | 550563 RRID:AB_393754 | Clone Meca-32 Mouse monoclonal Dilution 1:500 |
| Antibody | Secondary antibodies goat anti-rat Alexa Fluor-488 | Thermo Fisher Scientific, Invitrogen | Cat# A-11006 RRID:AB_2534074 | Goat anti-rat Clonality unknown Dilution 1:500 |
| Antibody | Secondary antibodies goat anti-chicken Alexa Fluor-568 | Thermo Fisher Scientific, Invitrogen | Cat# A-11041 RRID:AB_2534098 | Goat anti-chicken Clonality unknown Dilution 1:500 |
| Software | Imaris | http://www.bitplane.com/imaris/imaris | RRID:SCR_007370 | |

*Continued on next page*

*Appendix 1—key resources table continued*

| Reagent type (species) or resource | Designation | Source or reference | Identifiers | Additional information |
|---|---|---|---|---|
| Antibody | Anti-mouse CD62L | BioXcell | Cat# BE0021 RRID:AB_1107665 | This antibody reacts with L-selectin that is expressed by neutrophils, monocytes, and the majority of T- and B cells- Mouse monoclonal Dilution 1:500 |
| Antibody | Anti-CD19 | BD Biosciences | Cat# 552854 RRID:AB_394495 | Rat monoclonal antibody Dilution 1:200 |
| Enzyme-free cell dissociation buffer | Cell dissociation buffer | Thermo Fisher–Gibco | Cat# 13150016 | Dissociation of mammalian cells |
| Recombinant human fibronectin fragment | RetroNectin | Takara | Cat# T202 | This reagent promotes co-localization of lenti or retrovirus with the target cells and promotes the transduction efficiency |
| Antibody | Anti-CD169 | eBioscience | Cat# 12-5755-82 RRID:AB_2572625 | This antibody specifically binds to CD169, the receptor expressed in subset of macrophages and plays an important role in cell-cell adhesion |
| Antibody | Anti-F4/80 | Thermo Fisher Scientific, eBioscience | Cat# 48-4801-82 RRID:AB_1548747 | Rat anti-mouse Dilution 1:200 |
| Protease inhibitor cocktail | Complete Protease inhibitor cocktail | Sigma-Aldrich, Roche | Cat# 11697498001 | Inhibits broad spectrum of proteases |
| ELISA | Mouse Ccl19/MIP-3 | R&D System | Cat# DY440 | |
| ELISA | Mouse Ccl21/6Kine | R&D System | Cat# AF457 | |
| Recombinant protein | Mouse anti-Ccr7 | R&D System | Cat# MAB3477 | |
| Recombinant protein | Mouse anti-Ccl19 | R&D System | Cat# 440-M3 | |
| Inhibitor | Ubiquitin-conjugating enzyme E2 N (UBE2N) inhibitor or Ubc13 | Sigma-Aldrich | Cat# 662107 | This inhibitor controls the biological activity of UbcH13 |
| Inhibitor | IRAK1 inhibitor | Sigma-Aldrich | Cat# I5409 | |
| Inhibitor | PS1145 dihydrochloride | Sigma-Aldrich | Cat# P6624 | IκB kinase inhibitor |
| Inhibitor | Monensin Solution 100× | Thermo Fisher Scientific, eBioscience | Cat# 00-4505-51 | Inhibitor of intracellular transport |
| Assay | TUNEL-TMR | Roche | Cat# 12156792910 | Terminal deoxynucleotidyl transferase (TdT)-mediated dUTP nick end labeling (TUNEL) assay to detect apoptotic cells labeled with tetra-methyl-rhodamine |

