## [Decision Letter]

**Acceptance summary:**

Your study characterises an intriguing response to LPS, as a model of inflammation, wherein bone marrow progenitor cells are released directly into the lymphatics to replenish myeloid cells in lymph nodes. These data represent a very thorough investigation of a novel biological phenomenon and therefore will lay a new framework and will likely be impactful to the fields of haematopiesis and immunology.

**Decision letter after peer review:**

Thank you for submitting your article "Systemic inflammation recruits fast-acting anti-inflammatory innate myeloid progenitors from BM into lymphatics" for consideration by *eLife*. Your article has been reviewed by 2 peer reviewers, and the evaluation has been overseen by a Reviewing Editor and Carla Rothlin as the Senior Editor. The following individual involved in review of your submission has agreed to reveal their identity: Katherine MacNamara (Reviewer #2).

Essential Revisions:

While it would be challenging to strengthen the human data, the murine studies are already the major component of the manuscript, allow for a thorough analysis of mechanism, and could be strengthened as suggested especially by Reviewer 2 (see specific comments below). Given that a very large body of very thorough work is presented here and that these are exceptional months, we all agreed that it is not advisable to set up specific experiments now. If data that address any of the questions below were already available within the group, they would fit perfectly in the revised version of the manuscript. If not, the wording of the manuscript should be modified as suggested.

1. Please assess the impact of CCR7 blockade on NK cells and T cell subsets. Are anti-inflammatory factors increased? Could this be known from other studies? It should be at least commented on.

2. Please better characterise DCs function and numbers following CCR7 blockade. Alternatively please soften the conclusions to include other possible outcomes.

3. Please address all the comments aimed at improving clarity and better frame the conclusions presented both regarding the points above and throughout the manuscript.

*Reviewer #1 (Recommendations for the authors):*

In this manuscript authors demonstrate that acute systemic inflammation induces a new system of rapid migration of granulocyte-macrophage progenitors and committed macrophage-dendritic progenitors but not other progenitors or stem cells from BM to lymphatic capillaries. This traffic is mediated by Ccl19/Cccr7 and is NfkB independent but Traf activation dependent. This type of trafficking is anti-inflammatory with promotion of early survival.

Specifically, authors work shows the traffic of DC-biased myeloid progenitors through direct transit from BM to bone lymphatic capillaries. This type of trafficking is highly activated in endotoxic inflammation. Giving LPS to mice results in massive mobilization of myeloid progenitors from the BM to lymph and retention in LN takes place. This happens rapidly and before the appearance of these cells in PB. This type pf LPS challenge induces Ccr7 expression on GMPs as well as secretion of CcL9 in the LN. Importantly, loss of CcL9 or neutralizing Ccr7 inhibits GMP/MDP migration to the LN and inflammation induce mortality.

The studies are well performed and the data supports the conclusions. The role of this signaling axis in the recruitment of GMPs/MDPs has not been investigated in this detail.

*Reviewer #2 (Recommendations for the authors):*

This manuscript by Serrano-Lopez et al., characterizes an intriguing response to LPS wherein bone marrow progenitors are released directly into the lymphatics. This rapid response is directed via chemokines and dependent on TLR-Traf6-dependent, NFkB-independent signaling. This is a very interesting study and reveals an underappreciated role for the lymphatics in responding to acute inflammatory stress and highlights a novel route of trafficking for hematopoietic progenitors that may be important in a variety of settings. These data represent a very thorough investigation of a novel biological phenomenon and therefore will likely be highly impactful to the fields of hematopoiesis and immunology. However, these studies also raise several questions and require additional experiments/data to validate the findings. At the same time, while a strength of this manuscript is the combination of both murine and human data, the arrangement of the data is somewhat confusing and raises some additional questions that need to be addressed.

1. Data presented in Figure 1 and Supplemental Figure 1 are intriguing, but also a bit confusing. The human data begs a number of questions: first, are there differences in absolute numbers (per LN sample) as the frequency may be influenced by the abundance of mature cells, particularly in lymphoma patients. Additional discussion is required. Second, the combined human and murine data in these figures is disorganized; the blood PMN (supplemental data) seems relevant to what is known about G-CSF-induced mobilization, and should be shown in the main figures. Third, while the starting point for these studies seems to have stemmed from observations in humans, these findings are the least robust. One suggestion is to incorporate the human data at the end of the manuscript to suggest translational relevance, and begin with the most intriguing observation (which is found in Figure 1 H and I). This seems like an excellent starting point for the whole manuscript.

2. One aspect of HSC/HSPC function that was not addressed but may be very important is that of proliferation, especially because TRAF6-dependent Akt activation can regulate cell proliferation. Mobilization of progenitors due to G-CSF is associated with progenitor and HSC proliferation, and therefore, could be an important contributor to the numbers of cells observed. First, does proliferation change in the absence of TRAF6? Second, is the number of HSPCs in the blood different in these (TRAF6-deficient) mice? The second question stems from wondering whether the trafficking pattern is different when migration to the lymph/LNs is mitigated (ie, do they all get mobilized to the blood by default?). As TRAF6-deficient cells have baseline differences (ie, CCR7 expression S5.G) it seems there may be a variety of dysfunctional elements in these cells. Is this known and can this be commented on? If it is not feasible to look at proliferation or if these data haven't been generated it seems raising this as a potential caveat would be good.

3. The mechanisms accounting for the early death in models where myeloid progenitor accumulation in the LNs are not carefully investigated and therefore the conclusion the authors make regarding the role of myeloid progenitor differentiation in LNs as an anti-inflammatory response to prevent death, seems premature.

a. Specifically, CCR7 blockade may impact a variety of cells, including lymphocytes and natural killer (NK) in the LNs. CCR7 signaling is critical for maintaining normal T cell trafficking and T regulatory cell populations in the lymph nodes which very well could contribute to outcomes in this model. Have these lymphocytes been evaluated? Is it known if there are decreases in factors such as IL-10 or TGFb, anti-inflammatory factors?

b. Dendritic cell function is not analyzed and the phenotype described is not sufficient to define anti-inflammatory DCs. At the same time it is unclear how DC numbers change in these models where migration is blocked. CFU assays are reported, but do the numbers of mature cells differ?

c. Even without the conclusion on death, the observations of the described trafficking pattern and replenishment of the lymph node DC populations is novel and incredibly interesting, so the suggestion is to simply soften these conclusions and discuss other possible outcomes.

---

## [Author Response]

Essential Revisions:While it would be challenging to strengthen the human data, the murine studies are already the major component of the manuscript, allow for a thorough analysis of mechanism, and could be strengthened as suggested especially by Reviewer 2 (see specific comments below). Given that a very large body of very thorough work is presented here and that these are exceptional months, we all agreed that it is not advisable to set up specific experiments now. If data that address any of the questions below were already available within the group, they would fit perfectly in the revised version of the manuscript. If not, the wording of the manuscript should be modified as suggested.1. Please assess the impact of CCR7 blockade on NK cells and T cell subsets. Are anti-inflammatory factors increased? Could this be known from other studies? It should be at least commented on.

We understand the concern raised by the reviewers. Accordingly, we have modified the title, removed the survival data on Ccr7-deficient mice treated with lethal or sublethal doses of LPS, and edited the manuscript as suggested in Abstract, Introduction, Results and Discussion sections. We have circumscribed our analysis to define the effect of anti-Ccr7 on preventing the migration of myeloid progenitors.

Regarding the role of other inflammatory cells, we apologize for not having provided a set of important data that were generated during our transwell migration experiments (Figures 4E-F). We analyzed the secretion levels of chemokine inducer cytokines in the medium derived from lymph node cells. We found that LN cells did not secrete detectable levels of IFN-γ and the deficiency of Traf6 did not significantly modify the levels of secreted IFN-γ in BM cells (new (Figure 4—figure supplement 1B). Since IFN-γ is a major cytokine effector produced by activated NK cells, we believe that our data strongly suggest that NK cell activation may not be relevant to explain the migration of BM myeloid progenitors towards LN cells.

Similarly, the effect of Traf6 expression on non cell-autonomous, LN-mediated migration did not depend on the secretion of IL-1α, IL-2, IL-13, IL-4, TNF-α or IL-10 regulatory cytokines of inflammatory processes (Figure 4—figure supplements 1C-H). Together, this data re-inforce our overall set of data suggesting that inflammatory NK, T cells and B cells may not be critical to explain the rapid migration of myeloid progenitors from BM towards LN.

This data has been included in the Results and Discussion sections.

2. Please better characterise DCs function and numbers following CCR7 blockade. Alternatively please soften the conclusions to include other possible outcomes.

We have modified the title of the manuscript. We have also removed the survival data on Ccr7deficient mice treated with lethal or sublethal doses of LPS, and edited the manuscript as suggested in Abstract, Introduction, Results and Discussion sections. We have circumscribed our analysis to define the effect of anti-Ccr7 on preventing the migration of myeloid progenitors.

Our characterization of cDC derived from BM myeloid progenitors includes Gr1, CD11b and CD11c, which allow the identification of generic lymphoid-tissue cDCs. We have revised our language in the Results section to ensure we did not overstate our findings.

3. Please address all the comments aimed at improving clarity and better frame the conclusions presented both regarding the points above and throughout the manuscript.

As suggested, we have reviewed the manuscript to ensure that the clarity and conclusions support the data presented.

Reviewer #1 (Recommendations for the authors):In this manuscript authors demonstrate that acute systemic inflammation induces a new system of rapid migration of granulocyte-macrophage progenitors and committed macrophage-dendritic progenitors but not other progenitors or stem cells from BM to lymphatic capillaries. This traffic is mediated by Ccl19/Cccr7 and is NfkB independent but Traf activation dependent. This type of trafficking is anti-inflammatory with promotion of early survival.Specifically, authors work shows the traffic of DC-biased myeloid progenitors through direct transit from BM to bone lymphatic capillaries. This type of trafficking is highly activated in endotoxic inflammation. Giving LPS to mice results in massive mobilization of myeloid progenitors from the BM to lymph and retention in LN takes place. This happens rapidly and before the appearance of these cells in PB. This type pf LPS challenge induces Ccr7 expression on GMPs as well as secretion of CcL9 in the LN. Importantly, loss of CcL9 or neutralizing Ccr7 inhibits GMP/MDP migration to the LN and inflammation induce mortality.The studies are well performed and the data supports the conclusions. The role of this signaling axis in the recruitment of GMPs/MDPs has not been investigated in this detail.

We want to thank the reviewer #1 for his/her careful analysis of the manuscript and his/her kind words complimenting our work and stressing the fact that the role of this signaling axis in the recruitment of GMPs/MDPs had not been investigated in this detail in the literature. We have added a set of new data responding to specific concerns raised by both reviewers (see response to editors and reviewer #2 for specifics).

Reviewer #2 (Recommendations for the authors):This manuscript by Serrano-Lopez et al., characterizes an intriguing response to LPS wherein bone marrow progenitors are released directly into the lymphatics. This rapid response is directed via chemokines and dependent on TLR-Traf6-dependent, NFkB-independent signaling. This is a very interesting study and reveals an underappreciated role for the lymphatics in responding to acute inflammatory stress and highlights a novel route of trafficking for hematopoietic progenitors that may be important in a variety of settings. These data represent a very thorough investigation of a novel biological phenomenon and therefore will likely be highly impactful to the fields of hematopoiesis and immunology. However, these studies also raise several questions and require additional experiments/data to validate the findings. At the same time, while a strength of this manuscript is the combination of both murine and human data, the arrangement of the data is somewhat confusing and raises some additional questions that need to be addressed.1. Data presented in Figure 1 and Supplemental Figure 1 are intriguing, but also a bit confusing. The human data begs a number of questions: first, are there differences in absolute numbers (per LN sample) as the frequency may be influenced by the abundance of mature cells, particularly in lymphoma patients. Additional discussion is required.

We did not quantify the absolute number of SP cells in LN since this was essentially impossible to determine given the nature of the experimental design in the clinical trial. The size and origin of these LNs was diverse as presented in Figure 1A and figure1-supplement figure 1B. Unlike in the mouse model where complete regional LN chains were analyzed, the LNs sampled in the human trial did not include enough LN material to make the proposed quantification reliable to determine the overall cellularity of the regional lymph nodes.

Second, the combined human and murine data in these figures is disorganized; the blood PMN (supplemental data) seems relevant to what is known about G-CSF-induced mobilization, and should be shown in the main figures.

As suggested by the reviewer, we have included peripheral blood PMN counts in the Figure 1D.

Third, while the starting point for these studies seems to have stemmed from observations in humans, these findings are the least robust. One suggestion is to incorporate the human data at the end of the manuscript to suggest translational relevance, and begin with the most intriguing observation (which is found in Figure 1 H and I). This seems like an excellent starting point for the whole manuscript.

Regarding the recommended change in the order in the presentation of the human trial data, we respectfully disagree with the reviewer’s recommendation. If we had known the murine model results beforehand (which were surprising), we would have designed the human trial in the context of acute gram negative bacterial sepsis. Therefore, we have kept the human clinical study at the beginning of the manuscript (Figure 1A and figure 1-supplement figure 1A-D). We feel that to present the clinical data at the end of the manuscript would be difficult to justify and be presented logically as a translational consequence of our results. We also feel this manuscript faithfully reflects a process of bedside-to-bench rather than bench-to-bedside translation.

2. One aspect of HSC/HSPC function that was not addressed but may be very important is that of proliferation, especially because TRAF6-dependent Akt activation can regulate cell proliferation. Mobilization of progenitors due to G-CSF is associated with progenitor and HSC proliferation, and therefore, could be an important contributor to the numbers of cells observed. First, does proliferation change in the absence of TRAF6?

While LPS is inducing significant proliferation of myeloid progenitors (LK and LSK) after 12 hours post-administration but not at 4 hours (Zhao J et al., Cell Stem Cell 2014), proliferation seems not to be playing a major role at times of 1-3 hours after LPS challenge. The effect of Traf6 on proliferation of constitutively deficient hematopoietic stem cells and progenitors has already been published by our group (Fang J et al., Cell Reports, 2018). This information has been included in the Results and Discussion sections.

Second, is the number of HSPCs in the blood different in these (TRAF6-deficient) mice? The second question stems from wondering whether the trafficking pattern is different when migration to the lymph/LNs is mitigated (ie, do they all get mobilized to the blood by default?). As TRAF6-deficient cells have baseline differences (ie, CCR7 expression S5.G) it seems there may be a variety of dysfunctional elements in these cells. Is this known and can this be commented on? If it is not feasible to look at proliferation or if these data haven't been generated it seems raising this as a potential caveat would be good.

Given our findings presented in Figures 1C-J and figure 1-supplement figures 1E-N, we focused on the migration to LN and we did not attempt to define mechanisms of migration of myeloid progenitors to blood induced by LPS since this process has been well characterized in the literature. Also, the fact that this migration peaks at 6 hours after LPS administration further complicates a mechanistic analysis as the one provided by our approaches to the migration to lymphatics/LN within the first 3 hours after LPS administration. In any event, to explain this point, we are including in Author response image 1 a summary of the effects of Traf6 deficiency and neutralizing anti-Ccr7 and anti-Ccl19 on the migration of progenitors to peripheral blood by ZT7 (3 hours after LPS administration). We found no significant differences in the levels of migration at ZT7 amongst the different groups analyzed.

**Author response image 1. sa2fig1:** Peripheral blood count of myeloid progenitors at ZT7 (3 hours after PBS or LPS administration). A. Counts in Wt vs Traf6-deficient mice. B. Counts in Wt mice treated with either an isotype control (IgG2a) or neutralizing anti-Ccr7 antibody. C. Counts in Wt mice treated with an isotype control (IgG2a) or anti-Ccl19 neutralizing antibody. No difference is statistically significant.

3. The mechanisms accounting for the early death in models where myeloid progenitor accumulation in the LNs are not carefully investigated and therefore the conclusion the authors make regarding the role of myeloid progenitor differentiation in LNs as an anti-inflammatory response to prevent death, seems premature.a. Specifically, CCR7 blockade may impact a variety of cells, including lymphocytes and natural killer (NK) in the LNs. CCR7 signaling is critical for maintaining normal T cell trafficking and T regulatory cell populations in the lymph nodes which very well could contribute to outcomes in this model. Have these lymphocytes been evaluated? Is it known if there are decreases in factors such as IL-10 or TGFb, anti-inflammatory factors?

We understand the concern raised by the reviewer #2. Accordingly, we have modified the title, removed the survival data on Ccr7-deficient mice treated with lethal or sublethal doses of LPS, and edited the manuscript as suggested in Abstract, Introduction, Results and Discussion sections. We have circumscribed our analysis to define the effect of anti-Ccr7 on preventing the migration of myeloid progenitors.

Regarding the role of other inflammatory cells, we apologize for not having provided a set of important data that were generated during our transwell migration experiments (Figures 4E-F). We analyzed the secretion levels of chemokine inducer cytokines in the medium derived from lymph node cells. We found that LN cells did not secrete detectable levels of IFN-γ and the deficiency of Traf6 did not significantly modify the levels of secreted IFN-γ in BM cells (new Figure 4-supplement figure 1B). Since IFN-γ is a major cytokine effector produced by activated NK cells, we believe that our data strongly suggest that NK cell activation may not be relevant to explain the migration of BM myeloid progenitors towards LN cells.

Similarly, the effect of Traf6 expression on non cell-autonomous, LN-mediated migration did not depend on the secretion of IL-1α, IL-2, IL-13, IL-4, TNF-α or IL-10 regulatory cytokines of inflammatory processes (Figure 4-supplement figures 1C-H). Together, this data re-inforce our overall set of data suggesting that inflammatory NK, T cells and B cells may not be critical to explain the rapid migration of myeloid progenitors from BM towards LN.

This data has been included in the Results and Discussion sections.

b. Dendritic cell function is not analyzed and the phenotype described is not sufficient to define anti-inflammatory DCs. At the same time it is unclear how DC numbers change in these models where migration is blocked. CFU assays are reported, but do the numbers of mature cells differ?

We have removed any claim of anti-inflammatory role of the cDC2 found to express Ccl19 in LN. As indicated earlier, we have not analyzed changes in DC numbers and focused on progenitor counts. We have modified the Results and Discussion sections accordingly.

c. Even without the conclusion on death, the observations of the described trafficking pattern and replenishment of the lymph node DC populations is novel and incredibly interesting, so the suggestion is to simply soften these conclusions and discuss other possible outcomes.

We want to thank the reviewer #2 for his/her kind words and appreciation of the quality of our manuscript. We have softened the statements and conclusions as recommended.